# Transferring Linear Features Across Language Models With Model Stitching

**Alan Chen**
Brown University
alan_chen1@brown.edu

**Jack Merullo**
Goodfire
jack@goodfire.ai

**Alessandro Stolfo**
ETH Zürich
stolfoa@ethz.ch

**Ellie Pavlick**
Brown University
ellie_pavlick@brown.edu

## Abstract

In this work, we demonstrate that affine mappings between residual streams of language models is a cheap way to effectively transfer represented features between models. We apply this technique to transfer the *weights* of Sparse Autoencoders (SAEs) between models of different sizes to compare their representations. We find that small and large models learn similar representation spaces, which motivates training expensive components like SAEs on a smaller model and transferring to a larger model at a FLOPs savings. In particular, using a small-to-large transferred SAE as initialization can lead to 50% cheaper training runs when training SAEs on larger models. Next, we show that transferred probes and steering vectors can effectively recover ground truth performance. Finally, we dive deeper into feature-level transferability, finding that semantic and structural features transfer noticeably differently while specific classes of functional features have their roles faithfully mapped. Overall, our findings illustrate similarities and differences in the linear representation spaces of small and large models and demonstrate a method for improving the training efficiency of SAEs.

## 1 Introduction

Large language models (LLMs) have continually proven to be surprisingly intricate next token predictors that become more complex yet predictably better with scale [Kaplan et al., 2020], leaving a desire to characterize and *explain* the LLM's computations [Saphra and Wiegreffe, 2024, Olah et al., 2020]. Despite increasing efforts to study the rich internal mechanisms and representations of LLMs, the full computations still remain opaque [Sharkey et al., 2025, Engels et al., 2024].

A popular perspective to view the internal computation of LLMs is through the lens of features, or functions of the input that serve a downstream purpose like predicting the next token or composing to form model circuits [Olah et al., 2020, Huben et al., 2024]. The **Linear Representation Hypothesis** (LRH) posits that these features are represented as directions in a high dimensional space [Park et al., 2024, Elhage et al., 2022]. However, the features had been empirically observed to appear in superposition (activating with interference), making interpretation of residual stream activations difficult. Sparse dictionary learning methods [Olshausen and Field, 1996, Faruqui et al., 2015, Serre, 2006] such as Sparse Autoencoders (SAEs) have recently gained popularity for interpreting models through this lens by disentangling dense representations into sparsely activating feature sets [Huben et al., 2024, Bricken et al., 2023, Gao et al., 2025, Kissane et al., 2024]. A second hypothesis, **Strong Model Universality** [Li et al., 2015], predicts that good models learn the same data representations.

39th Conference on Neural Information Processing Systems (NeurIPS 2025).

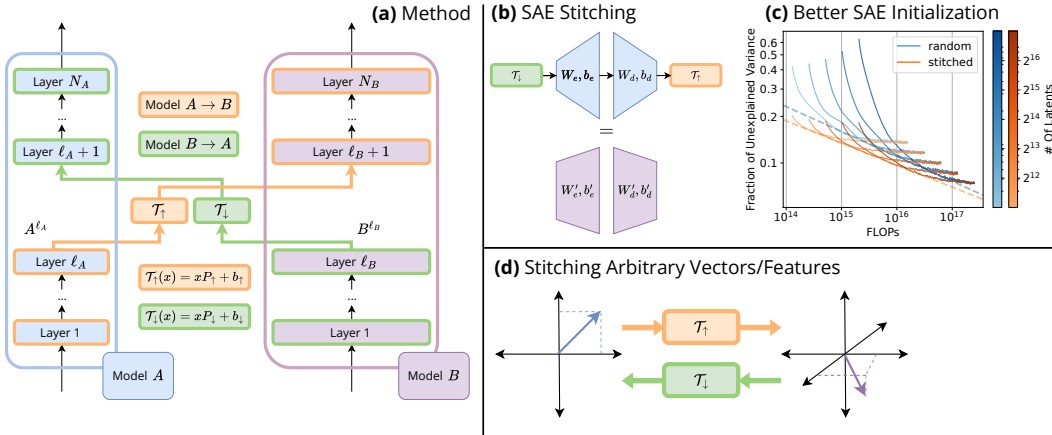

Figure 1: Overview of the main methodologies. **(a)** We train two affine mappings $\mathcal{T}_{\uparrow,\downarrow}$ concurrently to map between the residual streams of two language models $A$ and $B$. The mappings $\mathcal{T}_{\uparrow,\downarrow}$ are then used to transfer **(b)** the weights of entire SAEs from $A$ to $B$, which **(c)** give better initializations that save compute when training SAEs on $B$. The approximate "scaling law" for SAE training is shifted to the left when training from the transferred initialization, capturing the intuition that the transferred initialization saves the work of relearning shared features. More generally, the stitches can be used to transfer **(d)** arbitrary vectors (probes, steering vectors) between the residual stream spaces.

While it is unlikely that the strongest versions of these hypotheses faithfully reflect how these models work (that is, every feature is represented as a direction and every feature should be represented universally across good models) [Engels et al., 2025b, Csordás et al., 2024, Wei et al., 2022], there is significant evidence to support weaker versions of both hypotheses: linear features have been used to probe and intervene on model representations [Zou et al., 2023, Panickssery et al., 2023, i.a.] and are the fundamental idea underlying SAEs. One form of evidence for universality is model stitching, which learns a transformation mapping the latent spaces of two models [Lenc and Vedaldi, 2015, Bansal et al., 2021]. If the transformation is sufficiently simple, then it is claimed that the models encode information similarly. Other evidence includes recent work in "model diffing" [Lindsey et al., 2024] using crosscoders, a variant of SAEs, and transferring SAEs between base and chat models [Kissane et al., 2024] to study the effects of post-training.

Our work relies on the ansatz that if two models represent enough features as directions in similarly organized spaces, we should be able to transfer interpretable linear features between them. We propose the use of model stitches as the mappings for learned representations between language models. Our contributions (Figure 1) can be summarized as follows:

1. We learn stitches to transfer SAEs (§3) trained on one model to another, which are used to initialize SAE training runs on larger models with those trained on smaller ones (§3.1), allowing for an 50% FLOPs savings. We also demonstrate that stitches can be used to transfer probes and steering vectors (§4).

2. The stitches are affine mappings trained on within-family model pairs in Pythia [Biderman et al., 2023], GPT2 [Radford et al., 2019], and Gemma-2 [Team et al., 2024]. We find downstream metrics are preserved by the mappings, supporting weak universality (§2).

3. We perform a fine-grained analysis on feature transfer and analyze how well semantic and structural features transfer using the stitch (§5.1). We also find SAE features representing well-known universal functional features and note their transferability (§5.2).

## 2   Language Model Stitching

We would like to learn a mapping from the residual stream of language model $A$ to model $B$ at some layers, such that computation can start in the layers of $A$ and finish in $B$ (see Figure 1a). As notation, let $\mathbf{LM}^{\ell}(t) \in \mathbb{R}^{d_{\mathbf{LM}}}$ denote the representation in model $\mathbf{LM}$ of input token(s) $t$ which has hidden state dimension $d_{\mathbf{LM}}$ at layer $\ell$. Model stitching learns a mapping $\mathcal{T}$ (the **stitch**) between two model-layer

Table 1: Model-layer pairs that we stitch between and the associated downstream next token CE losses on OpenWebText. All activations are taken before the layer's computation. We primarily stitch from middle residual layers but for specific use cases (e.g., steering) we stitch between $3/4$ of the depth. The losses are computed stitching from $A$ to $B$, $B$ to $A$, $A$ to $B$ back to $A$, and $B$ to $A$ back to $B$. We calculate relative losses to the model that comes last and color code with blue and purple for model $A$ and model $B$ respectively.

| Model-Layer Pair | $A$ | $B$ | $A \to B$ | $B \to A$ | $A \to B \to A$ | $B \to A \to B$ |
|---|---|---|---|---|---|---|
| $A$: pythia-70m-deduped.3 $B$: pythia-160m-deduped.4 | 3.60 | 3.14 | 3.99 (+27%) | 4.21 (+17%) | 3.62 (+<1%) | 3.29 (+4.7%) |
| $A$: gpt2-small.6 $B$: gpt2-medium.10 | 3.07 | 2.77 | 3.08 (+11%) | 3.35 (+9.1%) | 3.07 (+<1%) | 2.81 (+1.4%) |
| $A$: gemma-2-2b.20 $B$: gemma-2-9b.33 | 2.52 | 2.36 | 3.28 (+39%) | 2.73 (+8.3%) | 2.53 (+<1%) | 2.66 (+13%) |

pairs $(A, \ell_A)$ and $(B, \ell_B)$. Informally, we would like $\left[\mathcal{T} \circ A^{\ell_A}\right](t) \approx B^{\ell_B}(t)$ where $\circ$ is function composition. The existence of such a mapping $\mathcal{T}$ relies on at least weak universality, especially if $\mathcal{T}$ preserves the hypothesis classes of model $A$ and $B$. We are interested in the particular case when $A$ and $B$ are decoder-only language models from the same family (trained on the same data) but $d_B \geq d_A$. In this setup (visualized in Figure 1a), we will consider the *two* stitching mappings $\mathcal{T}_\uparrow$, which maps "up" from $A$ to $B$, and $\mathcal{T}_\downarrow$, which maps "down" from $B$ to $A$. Furthermore, we will assume that both $\mathcal{T}_\uparrow$ and $\mathcal{T}_\downarrow$ are affine transformations:

$$\mathcal{T}_\uparrow : \mathbb{R}^{d_A} \to \mathbb{R}^{d_B}, \quad h_A \mapsto h_A P_\uparrow + b_\uparrow, \tag{1}$$

$$\mathcal{T}_\downarrow : \mathbb{R}^{d_B} \to \mathbb{R}^{d_A}, \quad h_B \mapsto h_B P_\downarrow + b_\downarrow, \tag{2}$$

where $h_A$ and $h_B$ are activations from $A$ and $B$, $P_\uparrow \in \mathbb{R}^{d_A \times d_B}$, $b_\uparrow \in \mathbb{R}^{d_B}$, $P_\downarrow \in \mathbb{R}^{d_B \times d_A}$, and $b_\downarrow \in \mathbb{R}^{d_A}$. Ideally, we would like $\mathcal{T}_{\{\uparrow,\downarrow\}}$ to be faithful to the downstream objective (i.e., language modeling). In practice, training directly on the next-token prediction objective would involve backpropagating gradients over the back halves of $A$ and $B$, which is unnecessarily expensive. We find that the reconstruction mean squared error (MSE) is a sufficient training objective to align the models. Despite the dimensionality gap, we also still desire "almost"-invertible transformations i.e. $\mathcal{T}_\uparrow \circ \mathcal{T}_\downarrow$ should be close to an identity and vice versa. Therefore, we also introduce a regularization penalty that encourages $\mathcal{T}_\uparrow$ and $\mathcal{T}_\downarrow$ to invert each other with relative strength $\alpha$, finding that the penalty improves fidelity (ablation in §A.2) and has further motivation when we consider transferring SAEs in §3.1 and §B.1. Formally, we train $\mathcal{T}_\uparrow$ and $\mathcal{T}_\downarrow$ concurrently via the loss function on a token $t$

$$\mathcal{L}(t) = \text{MSE}\left(\left[\mathcal{T}_\uparrow \circ A^{\ell_A}\right](t), B^{\ell_B}(t)\right) + \text{MSE}\left(\left[\mathcal{T}_\downarrow \circ B^{\ell_B}\right](t), A^{\ell_A}(t)\right)$$
$$+ \alpha \text{MSE}\left(\left[\mathcal{T}_\downarrow \circ \mathcal{T}_\uparrow \circ A^{\ell_A}\right](t), A^{\ell_A}(t)\right) + \alpha \text{MSE}\left(\left[\mathcal{T}_\uparrow \circ \mathcal{T}_\downarrow \circ B^{\ell_B}\right](t), B^{\ell_B}(t)\right). \tag{3}$$

We always learn the mappings from a fixed residual stream layer of model $A$ to the layer in model $B$ that maximizes the average correlation from Singular Vector Canonical Correlation Analysis (SVCCA) across a small sample of model activations [Raghu et al., 2017] (more details in §A.1). Using this procedure, the model-layer pairs that we stitch between are shown in Table 1.

**Downstream Performance.** In Table 1, we compute the downstream fidelity of mapping from $A$ to $B$, mapping from $B$ to $A$, and mapping in both inverse directions. First, we observe that next-token prediction performance is close to ground truth, supporting a weak universality hypothesis that models trained on the same data share features despite differing scales. However, we also find that downstream performance is consistently bottlenecked by the worse (generally smaller) model $A$. This result suggests that the gap between how $A$ and $B$ use their latent spaces prevents linear stitching between $A$ and $B$ from recovering the full performance of $B$. Finally, the inverse operation can be nearly lossless despite the intrinsic dimension mismatch prohibiting perfect invertibility.

# 3 Transferring SAEs

Sparse autoencoders (SAEs) have recently resurfaced in interpretability as a method of decomposing residual stream representations $x \in \mathbb{R}^d$ from a language model **LM** at layer $\ell$ into sparse nonnegative feature activations $f(x;\theta) \in \mathbb{R}^M$, with $M \gg d$ and $\theta = (W_e, b_e, W_d, b_d)$ of the encoder matrix, encoder bias, decoder matrix, and decoder bias respectively that specify the affine transformations in the forward pass, given as

$$f(x;\theta) = \sigma(xW_e + b_e), \tag{4}$$

$$\mathbf{SAE}(x;\theta) = f(x;\theta)W_d + b_d = \sum_{i=1}^{M} f_i(x;\theta)W_{d,i} + b_d. \tag{5}$$

The columns of the encoder matrix are the encoding/detection directions of the features. The rows of the decoder matrix are the decoding/representation directions of the features that the SAE decomposes into. Feature $i$ *activates* on token $t$ if $f_i\left(\mathbf{LM}^\ell(t);\theta\right) > 0$. SAEs are trained to reconstruct $x$ while having sparse activations via an objective function consisting of MSE between $x$ and $\mathbf{SAE}(x;\theta)$ and, depending on the architecture, an $L_p$ regularization (usually with $p = 1$) on $f(x;\theta)$ with strength controlled via a parameter $\lambda > 0$, respectively. We use pretrained SAEs with TopK activation [Gao et al., 2025] for Pythia[1] and JumpReLU SAEs for Gemma [Lieberum et al., 2024] and train TopK SAEs from scratch on the GPT2 models using SAELens [Bloom et al., 2024].[2]

## 3.1 Efficient SAE Transfer

Equipped with the trained stitch transformations, we make a key observation that an SAE with parameters $\theta = (W_e, b_e, W_d, b_d)$ trained on layer $\ell_A$ in $A$ can be transferred to layer $\ell_B$ in $B$. This procedure is illustrated in Figure 1b - specifically, for a latent $h_B = B^{\ell_B}(t)$ in model $B$, we (1) transfer down using $T_\downarrow$, (2) apply the original SAE parameterized by $\theta$, and (3) transfer back up using $T_\uparrow$. Because all the transformations are affine, this forward computation exactly specifies a new SAE on model $B$ parameterized by

$$\theta' = (W_e', b_e', W_d', b_d') = (P_\downarrow W_e, b_\downarrow W_e + b_e, W_d P_\uparrow, b_d P_\uparrow + b_\uparrow). \tag{6}$$

This relationship is derived in detail in §B.1. The explicit formulas for $\theta'$ also provide geometric intuition for the parameters of $\mathcal{T}_{\{\uparrow,\downarrow\}}$. Indeed, $P_{\{\uparrow,\downarrow\}}$ can be viewed as directly transforming the linear feature spaces of $A$ and $B$ as they impact the feature vectors through $W_e'$ and $W_d'$. On the other hand, $b_{\{\uparrow,\downarrow\}}$ are responsible for adjusting the "position" of the feature decomposition as they only alter the biases $b_e'$ and $b_d'$.

We report evaluations of the transferred SAEs in §C, which are better than random but certainly worse than a fully trained SAE on $B$ (e.g., a Fraction Unexplained Variance [FUV] of 0.21 to 0.42 in Gemma 2B to 9B). The metrics serve as evidence of weak universality in the midst of a representational gap between $A$ and $B$. As a partial explanation, we note that the weight matrices of the transferred SAE are still rank $\leq d_A < d_B$ i.e., the SAE is only detecting and writing to features along a rank $d_A$ subspace embedded within the rank $d_B$ ambient embedding space of model $B$, contributing to their lackluster performance.

**Training from a Stitch is More Efficient.** After obtaining a transferred SAE on $B$, we investigate how much continued training is required to match the performance of an SAE fully trained on model $B$. This question leads to a key application of the stitch and implicitly weak universality: **the transferred SAE can be used to initialize a new SAE in model $B$, "saving" the compute of having to relearn features that transfer well**. Concretely, consider the problem of training one SAE for two models of different sizes in the same family [Lieberum et al., 2024]. The straightforward approach is to separately train a SAE from scratch on each model. However, we find that given a fully trained SAE on the small model, training a stitch and using the transferred small model SAE

---

[1] https://github.com/EleutherAI/sparsify

[2] Gao et al. [2025] releases pretrained TopK SAEs for GPT2. However, since these SAEs use a LayerNorm normalization scheme but our stitches are trained on unnormalized residual stream activations, the SAE transfer computation does not extend as easily.

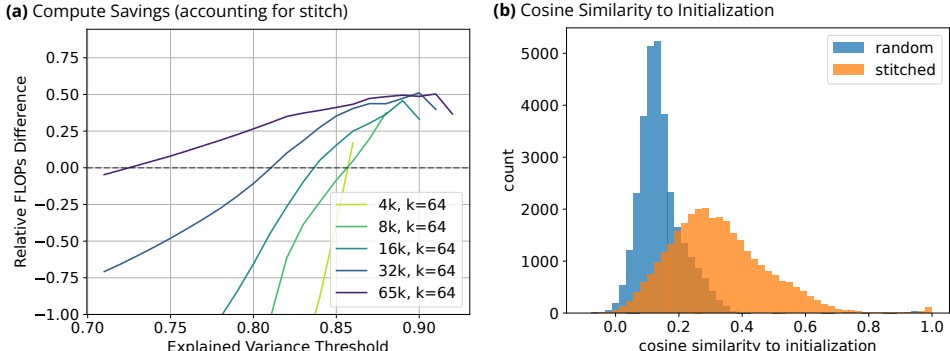

Figure 2: **(a)** In the Pythia model pair, transferred SAE initialization adjusted by the stitch FLOP count reaches explained variance thresholds in less FLOPs. For thresholds around $90\%$ explained variance, the moving average of explained variance of the SAE hits the threshold in around 30-50% less FLOPs. **(b)** Features when trained from stitched initialization have higher cosine similarity to their initial state than random initialization in the $M = 32768$ runs. Dead features are removed from consideration for clarity around $1.0$.

in the initialization scheme of the larger SAE allows for training of a comparable SAE to training from scratch but in a cheaper total FLoating point OPerations (FLOPs) budget. We support this claim by training TopK SAEs on the Pythia model pair and repeat with GPT2 in §D.1. We estimate FLOP counts based on the assumption that activations are cached prior to training both the stitch and the SAEs. All SAEs are trained using SAELens [Bloom et al., 2024] and other training and FLOP estimation details are included in §D.

Generally, we find that the cost of training the stitch on cached activations is much cheaper compared to training an SAE. Despite the cheap cost of the stitch, it allows us to stop the training of the SAE earlier when we use the transferred SAE as initialization. Figure 2a displays the relative FLOPs-to-hit-threshold savings of training from the transfer SAE initialization (accounting for the cost of training the stitch) vs. training from scratch initialization. More specifically, for the two initializations, we compare the number of FLOPs needed to get an SAE on Pythia-160m with explained variance above a particular threshold. At high explained variance values, we collect savings of between 30%-50% less FLOPs.

In Figure 1c we repeat the experiment for various latent sizes to create an approximate scaling law plot inspired by Gao et al. [2025]. We emphasize our fit is only an *approximation* meant for visualization as we estimate the law as a simple linear regression in log-log space instead of a proper scaling law (details and coefficients in §D.2) - we ignore the irreducible loss term because we are comparing the two laws against each other and stably fitting the term requires larger latent sizes to saturate the curve. Nonetheless, when training from stitch initialization, the fitted law is noticeably shifted to the left compared to random initialization, indicating that the stitch initialization reaches levels of reconstruction loss relatively faster consistently across the given latent sizes. For small latent sizes, such a procedure is less worth the additional compute of training the stitch, as adding in the compute required to train the stitch would result in the procedure becoming more expensive than just training from scratch. To confirm our intuition about the initialization, we also compute cosine similarities of decoder vectors at the final checkpoint to their original direction at initialization. In Figure 2b, we observe that the features trained from stitch initialization rotate less in aggregate than training from random, indicating the initialization is indeed placing features closer to their final values.

## 4   Downstream Applications

Despite SAEs not transferring without additional training, the mappings $\mathcal{T}_\uparrow$ and $\mathcal{T}_\downarrow$ are fundamentally general transformations that linearly relate the residual streams of two models. In this section, we examine the application of the stitch to transferring probes and steering vectors, which target specific vectors in the residual stream (Figure 1d). When applicable, following the intuition gained from Equation 6, we transfer feature vectors using just $P_{\uparrow,\downarrow}$ and do not include the bias.

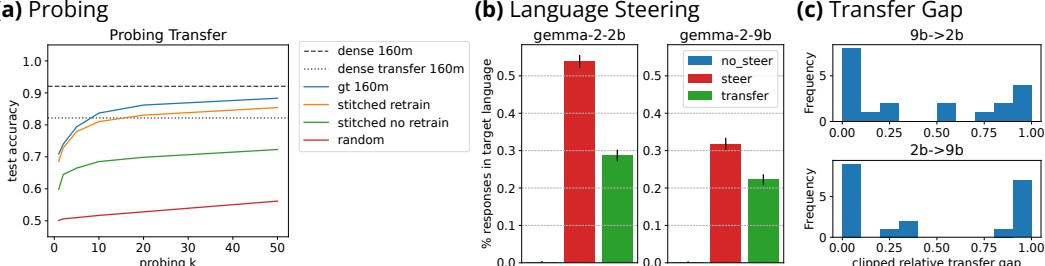

Figure 3: **(a)** Evaluations of transferred probes stitching from pythia-70m-deduped to pythia-160m-deduped averaged over 8 binary classification datasets. If the probe is retrained (orange), we almost recover ground truth performance (blue) across all probing $k$s and most datasets. Even if probe is not retrained, in most datasets we are able to probe significantly better than random (green). When directly probing on the residual stream, we find that transferring a probe trained on 70m-deduped (dotted) reaches similar accuracy to a probe trained on 160m-deduped (dashed) without retraining. **(b)** Response language steering vectors are able to be transferred between gemma-2-2b.20 and gemma-2-9b.33. From left to right, we chart the % of responses in the target language for no steering, ground truth steering, and steering using a transferred vector averaged over all languages $L$ in EuroParl and prompts. **(c)** The relative transfer gap distribution is bimodal with concentrations at 0 and 1, implying the transfer steering works well for some languages but poorly for others.

**Probing.** To begin, we consider the sparse probing task [Gurnee et al., 2023, Kantamneni et al., 2025, Karvonen et al., 2025] - specifically, using a certain subset of $k$ SAE features as probing features for a binary concept. The $k$ features are selected from the features found by a SAE and are selected via a simple max mean activation difference procedure: feature activations are collected over training samples, averaged over non-padding tokens, and the class-wise difference in means is computed. The $k$ features with largest difference in means are used as the probing features. Formally, let $\mathcal{D}_+$ and $\mathcal{D}_-$ be the set of activations at a layer on positive and negative training examples respectively. Then, we compute the set $\mathcal{F}_k$ of indices of the $k$ features as

$$\mathcal{F}_k = \text{argTopK}_i \left( \mathbb{E}_{x_+ \sim \mathcal{D}_+}[f_i(x_+; \theta)] - \mathbb{E}_{x_- \sim \mathcal{D}_-}[f_i(x_-; \theta)] \right). \tag{7}$$

In order to probe model $B$, we can transfer the SAE from $A$ to $B$ and use the activations from the same $k$ features in $\mathcal{F}_k$ to probe $B$. This procedure is equivalent to computing $\mathcal{T}_\downarrow(B^{\ell_B}(t))$, then applying a probe on $\mathcal{F}_k$'s activations on the stitched residual stream. Notably, the probe we use can either be completely reused as a pure evaluation of the stitch (i.e. the same probe that we trained when probing model $A$) or retrained to extract the full probing capability of the transferred features.

In Figure 3a, we present the test accuracies for probes averaged across 8 SAEBench datasets for $k = [1, 2, 5, 10, 20, 50]$. The results for all datasets presented separately is in §E. We compare against baselines of selecting features from a ground truth SAE, random features, and dense representations in model $B$. On average, we are able to probe successfully using the transferred features *without* retraining the probe - in particular, we do not collect activations from model $B$ outside of the data used to train the stitch. These results also hold with dense probes shown in the dashed and dotted lines: the dashed line is the skyline of training a probe on the dense representation of pythia-160m-deduped, whereas the dotted line is a probe trained on the dense representation of pythia-70m-deduped zero-shot applied to down-stitched residual streams from pythia-160m-deduped.

**Steering/Instruction Following.** In this section, we explore transferring steering vectors in a simple task: altering the response language from `en` to a target language $L$ in EuroParl [Koehn, 2005]. We compute a steering vector over the first 100 paired examples in each `en`-$L$ dataset by unit-normalizing the difference in mean activation on $L$ tokens vs. English tokens [Panickssery et al., 2023]. To steer, we clamp the component in the feature's direction to $\bar{z}$, where we set $\bar{z}$ to the mean of $\langle \mathbf{LM}^\ell(t), v \rangle$ over the tokens in $L$ (i.e., positive examples of the desired behavior). Explicitly, if $v$ is a steering vector and $\|v\|_2 = 1$, then we steer by modifying the hidden state at a token $t$ as

$$c = \bar{z} - \langle \mathbf{LM}^\ell(t), v \rangle, \tag{8}$$

$$\mathbf{LM}^\ell(t)' = \mathbf{LM}^\ell(t) + cv. \tag{9}$$

$v$ is transferred by computing $vP_{\{\uparrow,\downarrow\}}$, renormalizing, and recomputing $\bar{z}$ in the new model. We evaluate over a subset of 163 prompts from the IFEval dataset with the instructions stripped and aggregate the proportion of responses in the target language [Zhou et al., 2023, Stolfo et al., 2025]. Importantly, none of the prompts explicitly request the response to be in the target language.

We use the Gemma model pair and present the results in Figure 3b, where we plot the % of responses in the target language depending on the type of intervention: no steering, steering with a vector learned from the current model, and steering with a transferred vector learned from the other model. Averaged over $L$, we find that the transferred steering vector identifies a direction that successfully steers the model toward responding in the target language without explicit prompt instruction to do so. We break down the overall accuracy into individual language pairs (en, $L$) and find the transferred steering vector works well for some languages but not for others by defining a *clipped relative transfer gap* as the ratio of transfer steering performance to ground truth steering performance clipped to $[0, 1]$ for visualization (Figure 3c). We also note a positive correlation between language frequency and steering transfer effectiveness (§F.1). Although language steering appears to work somewhat well, we find that steering for general format instruction following has weaker results (§F.2).

## 5 Feature Analysis

Finally, we can gain intuition into specifically what features transfer well under the stitch. To this end, we consider a correlation-based metric that approximates *activation* and *downstream effect* similarity Bricken et al. [2023]. In particular, we compute the Pearson correlation between **attribution scores** induced by the logit weights and the model's next token predictions. Let $\text{Corr}(\cdot, \cdot)$ denote the Pearson correlation operator over data $x$ and let $v_i$ and $v_i'$ denote the logit weights given by unembedding the $i$th decoder vectors from $W_d$ and $W_d'$ respectively. The attribution correlation is given by

$$\text{Corr}_{t_\tau} \left( f_i \left( A^{\ell_A}(t_\tau); \theta \right) v_{i,t_{\tau+1},A}, f_i \left( B^{\ell_B}(t_\tau); \theta' \right) v'_{i,t_{\tau+1},B} \right) \tag{10}$$

where the correlation is computed over a set of tokens $t_\tau \in \mathcal{D}$ and $t_{\tau+1}$ is the next token. Intuitively, this score approximates the feature's importance in generating the next token prediction by taking the product of the activation and the downstream logit weight. We plot the histograms of attribution correlation scores against random SAEs in §G.

### 5.1 Semantic vs. Structural features

The first question we can pose is whether different types of features are transferred relatively differently according to the metric in Equation 10. The simplest classes of features are semantic features vs. structural/syntactic features. We outline a cheap but general experiment to generate this separation, visualized in Figure 4a with example classifications in Table 8. We first construct a synthetic dataset consisting of augmented versions of sentences drawn from a large text corpus. For each randomly drawn sentence, we use another LLM to generate $k$ augmented versions of the sentence with a prompt specifically instructing the LLM to ablate the semantic content of the sentence while maintaining the structure (§H.1). We then feed the original sentence and all augmented versions $(1 + k$ sentences in total) into the original language model and collect which SAE features activate on any token in the prompts. Features that activate consistently across the augmented versions of the same prompt are classified as **structural**, whereas all other features are classified as **semantic**. This classification is *stitch-agnostic* - we classify the features purely based on activations from the original SAE. As a concrete example, suppose we began with the prompt "I bought apples, bananas, and pears." We generate an augmented version of this prompt that preserves the structure of the sentence but ablates semantic content e.g. "John fostered cats, dogs, and fish." Consider a feature that activates on the last comma of a list: this feature would activate in both prompts, resulting in classification as a structural feature. However, a feature that activates on food or animals would only activate on one of the prompts or the other, resulting in classification as a semantic feature.

We plot the histograms of structural vs. semantic features with respect to the attribution correlation in Figure 4b for the GPT2 stitch, removing all dead features from consideration. Our main observation is that structural features tend to consistently transfer better, but semantic features are more polarized: they generally transfer well or they do not. Directly, this result implies that the stitch strongly transfers a group of semantic features, mostly transfers structural features, and leaves behind some semantic features. This experiment also supports that new dimensions in larger models could be

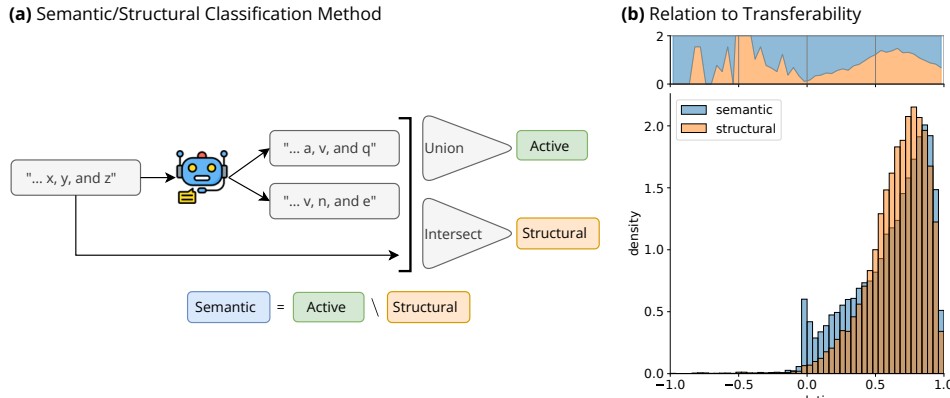

**(a)** Semantic/Structural Classification Method          **(b)** Relation to Transferability

Figure 4: **(a)** An overview of the feature analysis pipeline for a simple example where 2 augmentations are generated. Structural features activate on all prompts (intersection) whereas semantic features only activate on some but not all prompts. **(b)** The semantic/structural classification reveals a divergence in the attribution correlation transferability metric for non-dead features. We plot the densities of both categories separately and relative density above. Structural features transfer more consistently but semantic features are more polarized (dominate the upper and lower percentiles).

mostly allocated to developing increasingly specific semantic directions whereas the core language modeling spaces (which are more structural) remain similar.

## 5.2 Functional Features

Some of the structural features are related to known examples of universal features: in particular, entropy features and attention deactivation features [Gurnee et al., 2024]. We find that our stitch preserves the functional role of these features.

**Entropy Features.** Entropy features are characterized by large norm and high composition with the effective null space (we define as the bottom $2\%$ of singular values) of the unembedding matrix [Stolfo et al., 2024]. In gpt2-small, we find two features from an SAE that have these properties. Furthermore, we find that even after stitching, these properties still hold true (Figure 5)a, showing that the stitch does indeed transfer the functional role of these features.

**Attention Deactivation Features.** In order to identify attention deactivation neurons, we use the heuristic score and path patching techniques from Gurnee et al. [2024]. In particular, to identify potential candidates for composition of a feature with a downstream attention head, we compute the heuristic score between all features and downstream attention heads. Then, to test a relationship between a feature and attention head, we path zero-ablate the feature's contribution to a target attention head's input at the current token over a small token set and measure the change in the attention pattern to the <bos> token. We identify an attention deactivation feature in the gpt2-small SAE and find that it retains its role as a deactivation feature for a downstream head post-transfer to gpt2-medium (Figure 5b). Attention sinks have also been noted to have large components to the effective null space [Cancedda, 2024] and we find that is preserved as well (original: $0.27$, stitch: $0.77$). Somewhat mysteriously, we also observe the same feature acting as an attention *activation* feature in gpt2-medium after transfer (Figure 16).

## 6 Related Work

**Language Model Representations.** Language models have empirically been observed to learn vector representations of familiar human interpretable concepts like function application, truthfulness, factual knowledge, and refusal [Todd et al., 2024, Gurnee and Tegmark, 2024, Zou et al., 2023, Arditi et al., 2024]. Linear directions can also be useful for model editing, steering, and concept erasure [Li et al., 2023, Ilharco et al., 2023, Panickssery et al., 2023, Belrose et al., 2023]. [Huben et al., 2024, Bricken et al., 2023] reintroduced SAEs as a technique for decomposing residual streams

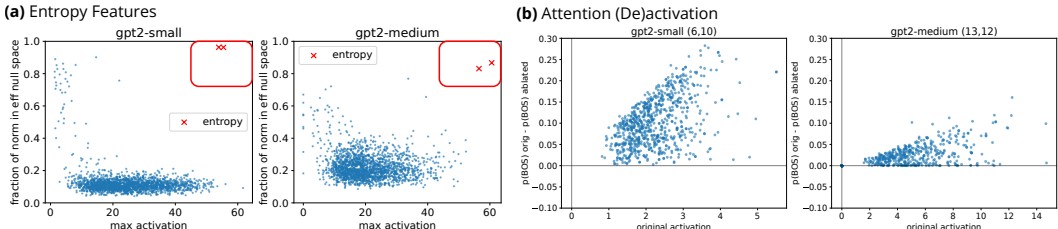

Figure 5: **(a)** Two entropy SAE features remain both large max activation and compose highly with the effective null space (bottom $2\%$ of singular values) before and after transfer. For clarity we take a randomly sampled subset of 2000 features. **(b)** Attention pattern on BOS token on path patching experiment. After transfer, we are still able to find a head such that zero ablating the contribution of the feature has results in decreased attention on the BOS token. We only plot tokens in which the original feature activates in gpt2-small.

which are hypothesized to geometrically superpose features [Elhage et al., 2022, Arora et al., 2018, Olah et al., 2020]. SAEs have been somewhat successfully applied to various downstream tasks like steering, probing, circuit analysis, and discovering interesting features [Karvonen et al., 2025, Ameisen et al., 2025, Ferrando et al., 2025, Marks et al., 2025]. Training SAEs has seen significant research effort, especially in architectural design and activation functions [Rajamanoharan et al., 2024, Gao et al., 2025, Bussmann et al., 2025]. Recently, however, limitations of SAEs have come into light and their use cases have become better understood [Paulo and Belrose, 2025, Chanin et al., 2025, Kantamneni et al., 2025, Engels et al., 2024]. In particular, a consistent assumption across many aforementioned works is the linearity of features [Mikolov et al., 2013, Park et al., 2024, Elhage et al., 2022]. However, recent work has demonstrated the existence of natural nonlinear features [Engels et al., 2025b, Csordás et al., 2024].

**Universal Representations Across LMs.** The intuition that well-generalizing models might all have similar representations has existed for quite some time [Li et al., 2015, Huh et al., 2024]. Model universality has been studied from the perspective of model stitching [Lenc and Vedaldi, 2015, Bansal et al., 2021, Jha et al., 2025, Csordás et al., 2025], representation similarity measures like SVCCA and CKA [Raghu et al., 2017, Kornblith et al., 2019, Barannikov et al., 2021], SAEs and variants [Lindsey et al., 2024, Lan et al., 2024], and feature/neuron/weight level analysis [Gurnee et al., 2024, Stolfo et al., 2024]. It has also been leveraged in adversarial attack literature [Zou et al., 2023]. Recently, transfer of steering vectors between models of different sizes using linear mappings between the final residual streams has also been explored in binary steering tasks and safety applications [Lee et al., 2025, Oozeer et al., 2025].

## 7 Conclusion

**Findings.** We find that we can faithfully stitch between language models of different sizes of the same family, which can be leveraged to transfer SAEs, probes, and steering vectors. The SAEs can be trained to convergence faster than training from scratch, demonstrating an application that uses weak universality to benefit SAE training. Probes and steering vectors can also be transferred with no additional training in specific cases. Finally, we perform a case study on GPT2 and find differences between how the stitch transfers semantic and structural features and discover that specific functional features have their roles preserved.

**Limitations and Future Work.** First, we only train the stitches on general internet text data between models in the same family with the same tokenizer. Natural follow ups are verifying the findings in a cross-family setting and finetuning the stitch to reconstruct unnaturally occurring special tokens or chat-templated data. The scaling laws are also theoretically incomplete because we lacked high-compute regime data to fit the irreducible loss. However, the transfer procedure is generalizable and other applications of the stitches to save compute or distill capabilities should be explored. For example, stitching could be used to transfer linear weight updates like LoRAs [Hu et al., 2022] which play nicely with the affine stitches and have already been established as similar to steering behavior [Engels et al., 2025a].

The semantic/structural classification is also imperfect - we only gain one type of distinction which is also sometimes noisy because the ablations are generated by another language model and transferability is purely correlational. A closer analysis of sensitivity of the stitching and semantic/structural methodology to different layers could also produce new insights into feature distributions. Finally, the probing and steering experiments deserve to be expanded upon to examine robustness as our experiments are still limited to particular tasks and settings.

## Acknowledgments

We would like to thank Michael Lepori, Jake Russin, and other members of the LUNAR Lab at Brown University for feedback at various stages of this project. We are also grateful to Neel Nanda for his valuable input during the early stages. AC would also like to thank Thomas Chang, Patrick Peng, and the CSCI2222 course at Brown for fruitful discussions about this work. AS acknowledges the support of armasuisse Science and Technology through a CYD Doctoral Fellowship. This project was supported in part by a Young Faculty Award from the Defense Advanced Research Projects Agency Grant #D24AP00261. Ellie Pavlick is a paid consultant for Google Deepmind. The content of this article does not necessarily reflect that of the US Government or of Google and no official endorsement of this work should be inferred.

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

# A  Stitch Training Details

The stitch training methodology is quite minimal. For the training dataset, we collect activations at the desired layers in both models over the first $180k$ samples of OpenWebText with a context size of $512$ ($128$ for Gemma due to compute constraints) and evaluate over the next 1k samples with the same context size. We mask out all special tokens. The stitches themselves are two separate Linear layers (with biases initialized to $0$) that map between the dimensions of $A$ and $B$. We use the Adam optimizer with a learning rate of `1e-4` and clip gradient norms to $1.0$. We found minimal sensitivity to learning rate schedule, so we just use a cosine annealing decay, and found that 2 epochs is sufficient for convergence (though even 1 is probably enough).

## A.1  Layer Selection

Suppose we have fixed some layer $\ell_A$ in model $A$ that we would like to stitch from. We determine the layer $\ell_B$ by computing a Singular Vector Canonical Correlation Analysis (SVCCA) over a small sample of tokens between the residual stream activations at $\ell_A$ and the candidate layer $\ell_B$ and taking the argmax. We use SVCCA because it is cheap to over a small set of activations while being directly related to how linearly related the two sets of activations accounting for noise in lower variance directions. We visualize all pairwise SVCCA values computed in Figure 6.

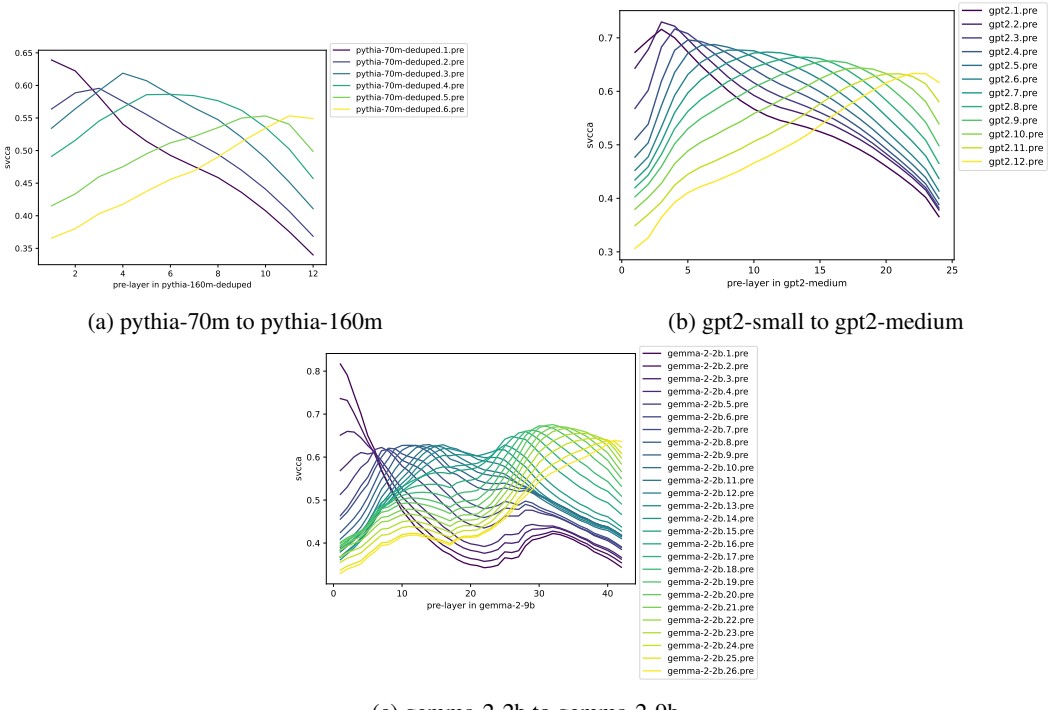

(a) pythia-70m to pythia-160m          (b) gpt2-small to gpt2-medium

(c) gemma-2-2b to gemma-2-9b

Figure 6: Computed SVCCA scores over all pairwise layers in the model pairs we stitch between. We end arbitrarily selecting a layer in the smaller model and choosing the layer in $B$ with the highest SVCCA coefficient.

## A.2  Inversion Ablation

See Figure 7.

# B  Transferring Details

Assume, we have two autoregressive language models $A$ and $B$ with latent embedding dimensions $d_A$ and $d_B$, respectively. Furthermore, assume there exist affine transformations $\mathcal{T}_\uparrow : (P_\uparrow, b_\uparrow) :$

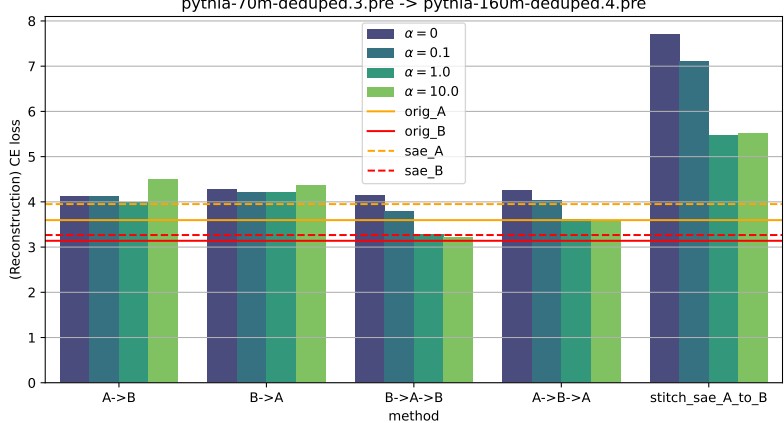

Figure 7: Strength of inversion ablation experiment. For various values of $\alpha$ we plot downstream next token CE loss for different transformations that should all be low - stitching from $A$ to $B$ and from $B$ to $A$, inversions, and the delta loss of an SAE transferred up from $A$ to $B$. We can see that $\alpha = 1.0$ strikes a good balance so we use it for all experiments.

$\mathbb{R}^{d_A \times d_B} \times \mathbb{R}^{d_B}$ and $\mathcal{T}_\downarrow = (P_\downarrow, b_\downarrow) : \mathbb{R}^{d_B \times d_A} \times \mathbb{R}^{d_A}$ as in Equation 1 that relate the residual streams of $A$ and $B$ at fixed layers in both models.

## B.1 SAE Methodology

Recall that a SAE : $\mathbb{R}^{d_A} \to \mathbb{R}^{d_A}$ defined on model $A$ can be written using the following computation:

$$\text{SAE}(x; \theta) = \sigma(xW_e + b_e)W_d + b_d, \tag{11}$$

parameterized by the 4-tuple $\theta = (W_e, b_e, W_d, b_d)$ and an activation function $\sigma$.

The insight is that we can think of transferring a sparse autoencoder from model $A$ to model $B$ as capturing the following computation. For every latent in model $B$ $h_B \in \mathbb{R}^{d_B}$:

1. Stitch $h_B$ to $A$ using $\mathcal{T}_\downarrow$.
2. Apply the SAE on the stitched latent.
3. Stitch the reconstruction back to $B$ using $\mathcal{T}_\uparrow$.

It turns out this computation can be collapsed into a reparameterized SAE on model $B$ because all of the transformations are affine.

$$
\begin{aligned}
(\mathcal{T}_\uparrow \circ \textbf{SAE} \circ \mathcal{T}_\downarrow)(h_B) &= \textbf{SAE}(h_B P_\downarrow + b_\downarrow; \theta)P_\uparrow + b_\uparrow \\
&= \Big(\sigma([h_B P_\downarrow + b_\downarrow]W_e + b_e)W_d + b_d\Big)P_\uparrow + b_\uparrow \\
&= \sigma(h_B[P_\downarrow W_e] + [b_\downarrow W_e + b_e])[W_d P_\uparrow] + [b_d P_\uparrow + b_\uparrow] \\
&= \textbf{SAE}(h_B; \theta'),
\end{aligned}
$$

where

$$\theta' = (P_\downarrow W_e, b_\downarrow W_e + b_e, W_d P_\uparrow, b_d P_\uparrow + b_\uparrow) \tag{12}$$

is another sparse autoencoder, but now mapping from $\mathbb{R}^{d_B} \to \mathbb{R}^{d_B}$. Remark: one slight caveat is that SAE$'$'s feature matrices are still only rank $d_A$.

Observe that the matrices $P_\uparrow$ and $P_\downarrow$ do the grunt work in manipulating the actual feature spaces, whereas the biases $b_\uparrow$ and $b_\downarrow$ are just relocating the "position" of the residual stream in space and do not adjust the feature spaces themselves.

We also notice that assuming $\textbf{SAE}(x; \theta) \approx x$, the reconstruction loss

$$\|\textbf{SAE}(h_B; \theta') - h_B\|_2 \tag{13}$$

is low when $\mathcal{T}_\uparrow$ and $\mathcal{T}_\downarrow$ invert each other (as expected). This inspires the inclusion of the additional inversion penalty in training the transformations $\mathcal{T}_\uparrow$ and $\mathcal{T}_\downarrow$ (Equation 3).

Table 2: Summary metrics of transferred SAEs (no training). Results are displayed as original (A) / transfer (B). $L_0$ evaluations are dependent on architecture - the Pythia and GPT2 SAEs are Top-K and Gemma are JumpReLU.

| Metric | pythia-70m.3 / pythia-160m.4 | gpt2-small.6 / gpt2-medium.10 | gemma-2-2b.20 / gemma-2-9b.33 |
|---|---|---|---|
| $L_0$ | 16.0 / 16.0 | 32.0 / 32.0 | 77.2 / 74.7 |
| FUV | 0.17 / 0.52 | 0.11 / 0.39 | 0.21 / 0.42 |
| Delta Loss | 0.36 / 2.16 | 0.08 / 0.96 | 0.50 / 0.85 |
| Dead Features % | 5.2% / 2.9% | <1% / < 1% | 1.5% / 3.5% |

## B.2 Probes and Steering Vectors

To transfer a linear probe from $A$ to $B$, we simply apply the probe on the down-stitched residual stream $\mathcal{T}_\downarrow(B^{\ell_B}(t))$. If the probe is linear and has normal direction $w$, it is equivalent to using the vector $wP_\downarrow^T$ in $B$. Steering vectors are just transferred directly using $P_\uparrow$ without the bias.

## C   Zero-Shot SAE Evaluations

As basic evaluations of how well the feature spaces match, we can compute some core metrics of our zero shot transferred SAEs and compare them against the original SAEs. We report $L_0$, Fraction of Unexplained Variance (FUV), Delta loss, and dead features $\%$ in Table 2. Importantly, we note that the transferred SAEs are not perfect - one explanation of why is that the weight matrices of the transferred SAE (Equation 6) are still rank $\leq d_A$.

## D   SAE Training and FLOP Estimation Details

All SAEs we train are TopK SAEs trained using SAELens on unnormalized residual stream activations. We train SAEs with latent sizes 4096, 8192, 16384, 32768, and 65536. We abide by the following practices:

1. We normalize the decoder vectors to unit norm each iteration.
2. When randomly initializing, we initialize the decoder and encoder as transposes of each other.
3. We use a constant learning rate schedule and just use $0.0001$ as the learning rate.
4. We do not use an auxiliary loss for ease of FLOPs estimation (discussed below).

SAELens includes the auxiliary loss for TopK training by default. In order to work around disabling it, we set the `dead_feature_window` parameter to an arbitrarily large number so that dead feature computations are never run during training.

We estimate FLOP counts of training the stitches and SAEs with cached activations by simply computing the FLOPs of one forward and backward pass for the desired module on a batch of dummy data inside of the context manager `torch.utils.flop_counter.FlopCounterMode`. We then scale by the number of training iterations. Example FLOP estimates for a set of SAEs is shown in 3. As a minor fact, we do find that with larger latent sizes, the number of dead features in the

Table 3: FLOPs estimates for various important procedures with caching activations. All SAEs are trained with sparsity $k = 64$ and width 32k. A full run is 4B tokens (120k iterations) for the SAEs and 200M (36k iterations) tokens for the stitch. The Pythia-70m SAE is trained post layer 2 and the Pythia-160m SAEs are trained post layer 3.

| Procedure | FLOPs (w/ caching) |
|---|---|
| Pythia-70m SAE (Scratch) | $8.2 \times 10^{16}$ |
| Pythia-160m SAE (Scratch & Transfer) | $1.2 \times 10^{17}$ |
| Stitching Layer | $1.4 \times 10^{15}$ |

*original* SAE (on $A$) increases since we are not using the auxiliary loss. Since dead features tend to

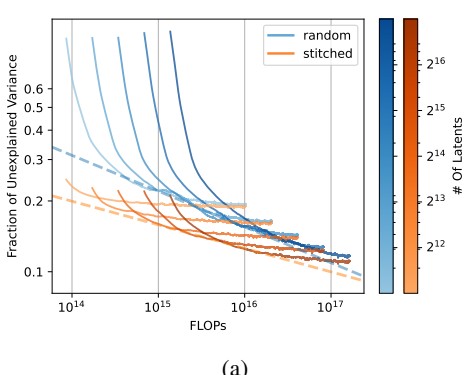 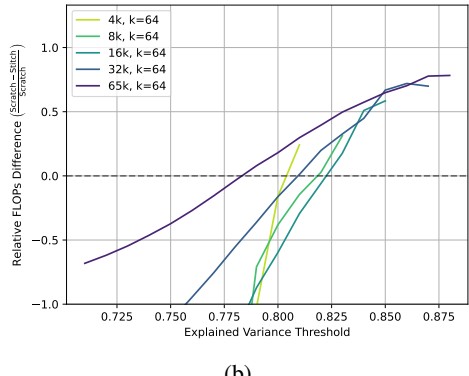

(a) (b)

Figure 8: (a) SAE "scaling law" for transfer results on GPT2 models with $k = 32$. (b) Relative FLOPs Difference between the initializations.

be inherited in the stitched SAE initialization, we end up with more dead features when training from the stitched SAE initialization. This could be remediated by reincluding the auxiliary loss (which would require careful FLOPs estimation), but we did not test this claim.

### D.1 GPT2 SAE Transfer

In Figure 8, we replicate the same plots as Figure 1c but for the GPT2 series models, with SAEs trained on 2B tokens and $k = 32$.

### D.2 Scaling Law Fit

We fit the scaling law by sampling the frontier MSE achieved at various FLOPs thresholds and fitting a linear model on the log-log scale (as opposed to fitting an irreducible loss term - unfortunately we do not have enough high-FLOPs points due to compute constraints to fit a complete scaling law stably). However, for our purposes, since theoretically the irreducible loss is the same for all initializations, so purely for relative comparison between two scaling laws it is not crucial. This results in a law of the form

$$L(C) \approx AC^{-\beta} \tag{14}$$

instead of

$$L(C) \approx L_\infty + AC^{-\beta} \tag{15}$$

where $L_\infty$ is the irreducible loss. We report the fitted coefficients for the approximate laws in Figure 1c (Pythia) and Figure 8a (GPT2) in Table 4.

Table 4: Approximate scaling law fitted coefficients for all laws we fit.

| Model/[Scratch, Stitch] | $A$ | $\beta$ |
|---|---|---|
| pythia-160m-deduped/scratch | 41.2 | 0.16 |
| pythia-160m-deduped/stitch | 18.5 | 0.14 |
| gpt2-medium/scratch | 42.3 | 0.15 |
| gpt2-medium/stitch | 5.0 | 0.10 |

## E Full Probing Results

See Figure 9 and Figure 10 for the fully decomposed probing results on each of the 8 SAEBench datasets.

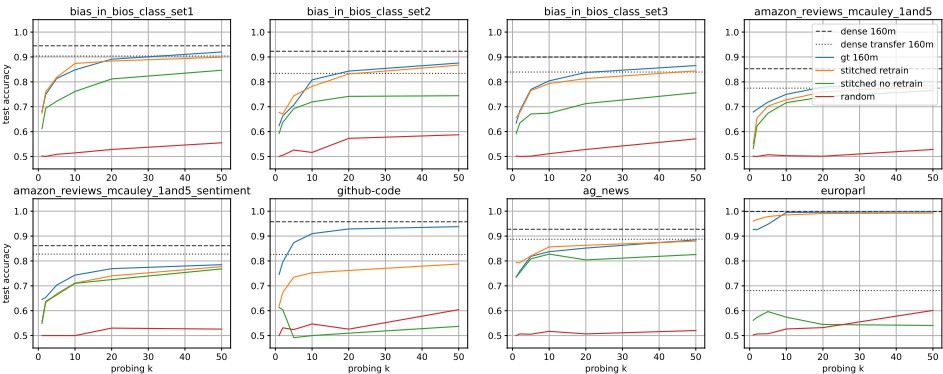

Figure 9: Transferring sparse probes from pythia-70m-deduped to pythia-160m-deduped works reasonably well in most datasets except code and language without retraining the probe on pythia-160m activations. Retraining the probe on the same features recovers ground truth performance.

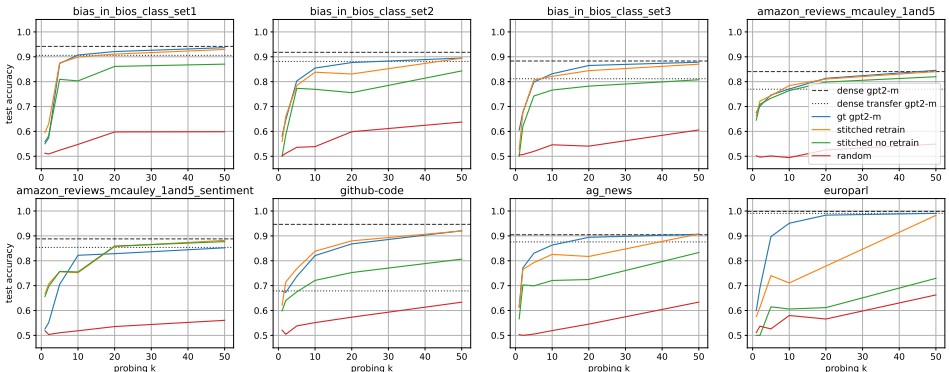

Figure 10: Sparse probing results on GPT2 SAEs trained from scratch. The conclusions are similar to the Pythia results - we can probe better than random using transferred probes. Code and language domains again seem hard to directly transfer but are fixed with retraining the probes.

## F    Steering Experiment Details

We do feature selection and coefficient fitting over the first 250 samples from Europarl for the dataset. To evaluate, we use `langdetect` to detect the most likely language of the response to the input prompt. We compute confidence intervals using Clopper-Pearson intervals at the $\alpha = 0.05$ level.

### F.1    Specific Language Results

We report the full accuracies for all languages in Tables 5 and 6.

The clipped relative transfer gap is defined as

$$\text{Clip}\left(\frac{\text{Tranfer Perf}}{\text{Ground Truth Perf}}, 0, 1\right), \tag{16}$$

and we define $0/0$ as $0$ and $\infty$ or $-\infty$ are clipped to $1$ and $0$ respectively.

We can compute a language frequency approximation by sentence-tokenizing a corpus of text and using `langdetect` to classify the language of each sentence, counting the frequencies. In Figure 11, we plot the clipped relative transfer gap against the language log frequency (computed over 100k samples of OpenWebText) and identify a positive correlation with frequent languages having better transfer performance.

Table 5: Full language steering results for gemma-2-2b. Format is [accuracy, (confidence interval low, confidence interval high)].

|  | No Steering (2b) | Steering (2b) | Transfer Steering (9b→2b) |
| --- | --- | --- | --- |
| **bg-en** | 0.00 (0.00, 0.02) | 0.31 (0.24, 0.38) | 0.01 (0.00, 0.03) |
| **cs-en** | 0.00 (0.00, 0.02) | 0.42 (0.34, 0.50) | 0.02 (0.00, 0.05) |
| **da-en** | 0.01 (0.00, 0.04) | 0.42 (0.34, 0.50) | 0.25 (0.18, 0.32) |
| **de-en** | 0.00 (0.00, 0.02) | 0.81 (0.74, 0.87) | 0.79 (0.71, 0.85) |
| **el-en** | 0.00 (0.00, 0.02) | 0.85 (0.79, 0.90) | 0.48 (0.41, 0.56) |
| **en-es** | 0.01 (0.00, 0.03) | 0.74 (0.66, 0.80) | 0.69 (0.61, 0.76) |
| **en-et** | 0.00 (0.00, 0.02) | 0.28 (0.21, 0.35) | 0.01 (0.00, 0.04) |
| **en-fi** | 0.00 (0.00, 0.02) | 0.40 (0.32, 0.48) | 0.06 (0.03, 0.10) |
| **en-fr** | 0.01 (0.00, 0.03) | 0.83 (0.76, 0.88) | 0.77 (0.69, 0.83) |
| **en-hu** | 0.01 (0.00, 0.03) | 0.72 (0.65, 0.79) | 0.06 (0.03, 0.10) |
| **en-it** | 0.00 (0.00, 0.02) | 0.69 (0.62, 0.76) | 0.60 (0.52, 0.67) |
| **en-lt** | 0.00 (0.00, 0.02) | 0.23 (0.17, 0.31) | 0.00 (0.00, 0.02) |
| **en-lv** | 0.00 (0.00, 0.02) | 0.50 (0.42, 0.58) | 0.00 (0.00, 0.02) |
| **en-nl** | 0.00 (0.00, 0.02) | 0.67 (0.59, 0.74) | 0.64 (0.57, 0.72) |
| **en-pl** | 0.01 (0.00, 0.03) | 0.62 (0.54, 0.69) | 0.18 (0.12, 0.25) |
| **en-pt** | 0.01 (0.00, 0.03) | 0.76 (0.69, 0.82) | 0.63 (0.55, 0.70) |
| **en-ro** | 0.00 (0.00, 0.02) | 0.40 (0.32, 0.48) | 0.09 (0.05, 0.15) |
| **en-sk** | 0.00 (0.00, 0.02) | 0.19 (0.13, 0.26) | 0.01 (0.00, 0.04) |
| **en-sl** | 0.00 (0.00, 0.02) | 0.28 (0.21, 0.35) | 0.02 (0.00, 0.05) |
| **en-sv** | 0.00 (0.00, 0.02) | 0.69 (0.62, 0.76) | 0.48 (0.41, 0.56) |

Table 6: Full language steering results for gemma-2-9b. Format is [accuracy, (confidence interval low, confidence interval high)].

|  | No Steering (9b) | Steering (9b) | Transfer Steering (2b→9b) |
| --- | --- | --- | --- |
| **bg-en** | 0.00 (0.00, 0.02) | 0.24 (0.18, 0.31) | 0.00 (0.00, 0.02) |
| **cs-en** | 0.00 (0.00, 0.02) | 0.55 (0.47, 0.63) | 0.00 (0.00, 0.02) |
| **da-en** | 0.00 (0.00, 0.02) | 0.65 (0.57, 0.72) | 0.14 (0.09, 0.20) |
| **de-en** | 0.00 (0.00, 0.02) | 0.85 (0.78, 0.90) | 0.83 (0.77, 0.89) |
| **el-en** | 0.00 (0.00, 0.02) | 0.87 (0.81, 0.92) | 0.04 (0.02, 0.09) |
| **en-es** | 0.01 (0.00, 0.03) | 0.80 (0.73, 0.86) | 0.69 (0.61, 0.76) |
| **en-et** | 0.00 (0.00, 0.02) | 0.36 (0.29, 0.44) | 0.00 (0.00, 0.02) |
| **en-fi** | 0.00 (0.00, 0.02) | 0.17 (0.12, 0.24) | 0.01 (0.00, 0.04) |
| **en-fr** | 0.01 (0.00, 0.03) | 0.09 (0.05, 0.15) | 0.69 (0.61, 0.76) |
| **en-hu** | 0.01 (0.00, 0.03) | 0.02 (0.00, 0.05) | 0.01 (0.00, 0.03) |
| **en-it** | 0.00 (0.00, 0.02) | 0.07 (0.04, 0.13) | 0.56 (0.48, 0.64) |
| **en-lt** | 0.00 (0.00, 0.02) | 0.00 (0.00, 0.02) | 0.00 (0.00, 0.02) |
| **en-lv** | 0.00 (0.00, 0.02) | 0.00 (0.00, 0.02) | 0.00 (0.00, 0.02) |
| **en-nl** | 0.00 (0.00, 0.02) | 0.09 (0.05, 0.14) | 0.53 (0.45, 0.61) |
| **en-pl** | 0.00 (0.00, 0.02) | 0.01 (0.00, 0.03) | 0.01 (0.00, 0.03) |
| **en-pt** | 0.01 (0.00, 0.03) | 0.09 (0.05, 0.14) | 0.62 (0.54, 0.69) |
| **en-ro** | 0.00 (0.00, 0.02) | 0.00 (0.00, 0.02) | 0.01 (0.00, 0.03) |
| **en-sk** | 0.00 (0.00, 0.02) | 0.40 (0.33, 0.48) | 0.01 (0.00, 0.03) |
| **en-sl** | 0.00 (0.00, 0.02) | 0.27 (0.20, 0.34) | 0.00 (0.00, 0.02) |
| **en-sv** | 0.00 (0.00, 0.02) | 0.79 (0.71, 0.85) | 0.31 (0.24, 0.39) |

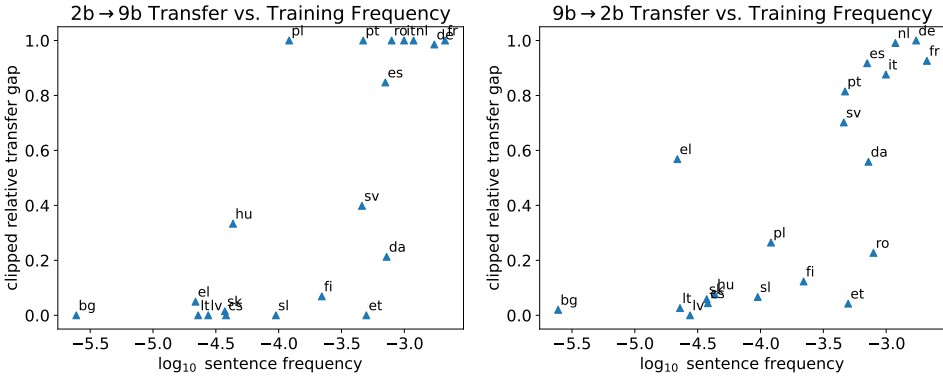

Figure 11: A positive correlation exists between being more frequent in natural language training data and having stronger transfer steering performance (relative to ground truth performance).

## F.2 Other Steering Experiments

We also trained a stitch between gemma-2-2b-it.20 and gemma-2-9b-it.33 and computed steering vectors for instruction following tasks directly following Stolfo et al. [2025]. The task setup involves responding to prompts without explicit instructions formatted using chat templating. We then apply a DiffMean steering vector at each token and verify whether the instruction is followed by the response. We found weaker steering results - in no case did transfer steering improve upon ground truth steering and small to big results are weak in general. In some cases, we find that ground truth steering performance is matched and generally stitched steering can still provide improvements over no steering whatsoever in some instructions.

Our current conjecture for why these results are much weaker than the response language results is that the steering vector is computed on the final token of the input prompt. Previous tokens in the input prompt include chat templating tokens like `<[start,end]_of_turn>`. The stitch is not trained on these tokens (they do not naturally occur in OpenWebText for example) and thus does not learn to properly reconstruct the important components of these tokens, which likely have very complex underlying latent representations but are important for steering. Furthermore, as is clear from the plots, the IFEval dataset is incredibly small so the confidence intervals are wide. The steering vectors are probably noisy for the instructions that have few examples (about 10 pairs) and the noise is compounded through transferring.

## G   Attribution Correlation Histograms

We plot histograms of the attribution correlation metric for our model-layer pairs in Figure 14. We plot the semantic/structural difference for Pythia and Gemma in Figure 15.

## H   Semantic / Structural Augmentation

### H.1   Prompt

We prompt `gemma-2-9b-it` 5 times with temperature 1.0 using the following prompt (generated using GPT-4 and mildly edited), resulting in 6 versions of the same sentence. We use context size 128.

```
Transform the given sentence so that its meaning is completely unrelated,
    but the syntax, punctuation, and grammatical structure remain identical.
     This means:
- Keep all function words (e.g., "the," "and," "while") and punctuation (e.
    g. commas, periods, brackets, parentheses, dashes) unchanged.
- Replace content words (nouns, verbs, adjectives, adverbs) with words from
     entirely different semantic domains (e.g., "run" -> "melt", "dog" -> "
    radio").
```

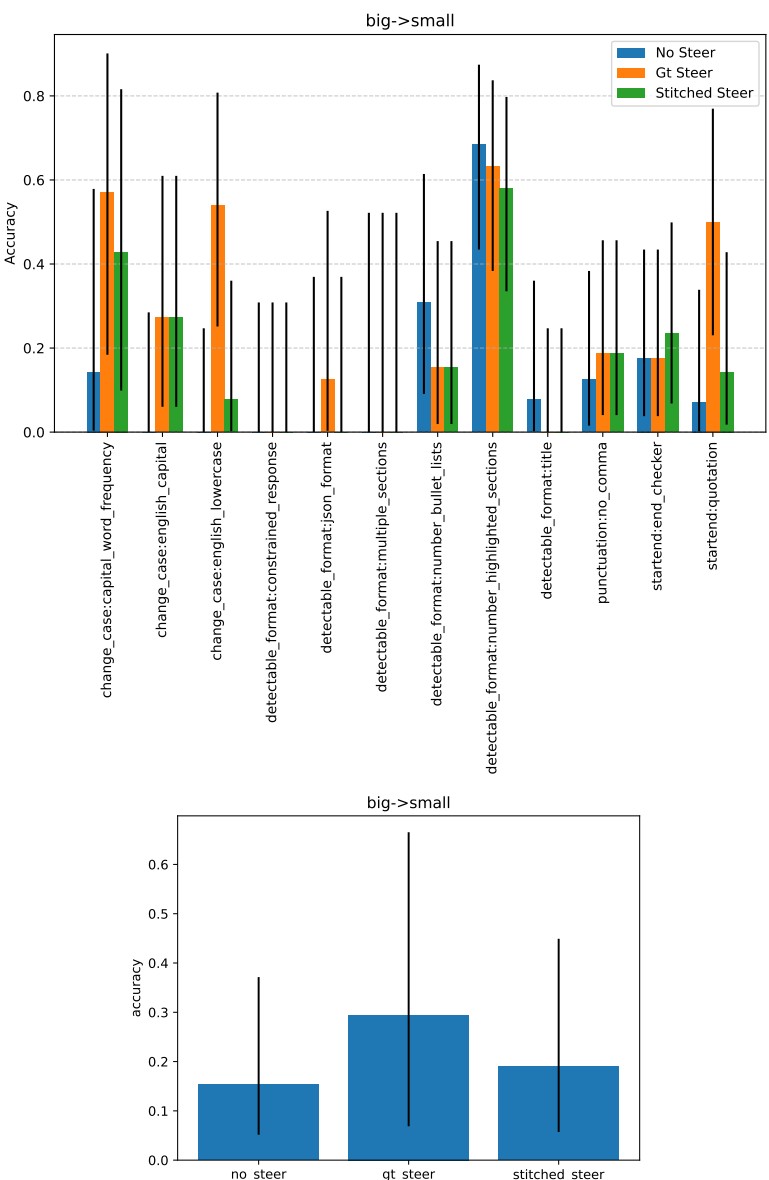

Figure 12: Big to small IFEval steering. We find that casing instructions are generally easiest to transfer.

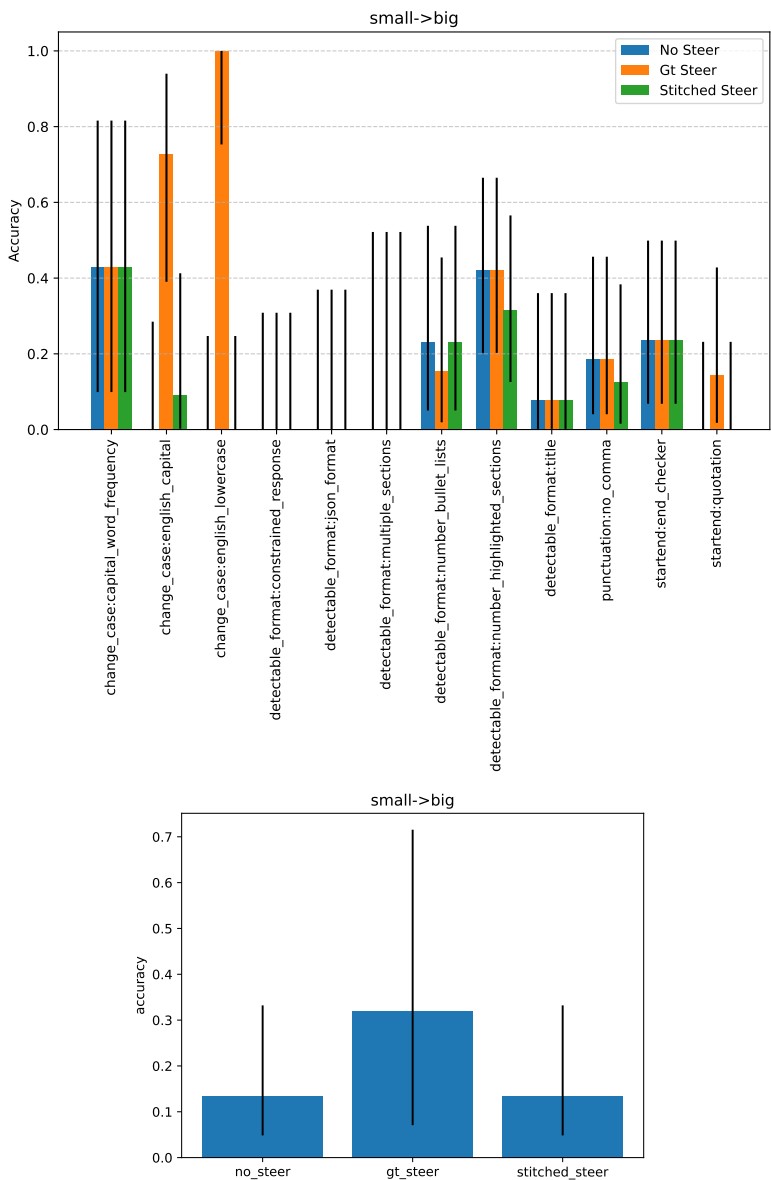

Figure 13: Small to big IFEval steering. Despite strong performance in ground truth steering in many cases, the transferred vectors are unable to identify the correct subspaces to induce the instruction following.

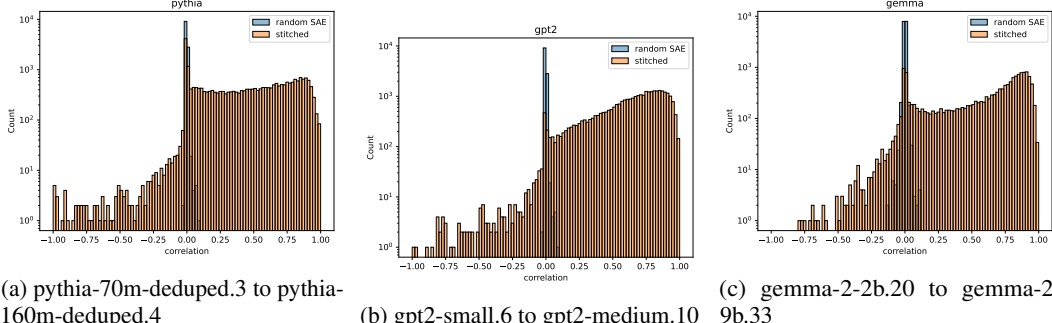

(a) pythia-70m-deduped.3 to pythia-160m-deduped.4

(b) gpt2-small.6 to gpt2-medium.10

(c) gemma-2-2b.20 to gemma-2-9b.33

Figure 14: Attribution correlation histograms for three model stitches. We can see that for most features we do much better than a random stitch. However, there are some straggling features that appear to get either flipped or do not transfer well.

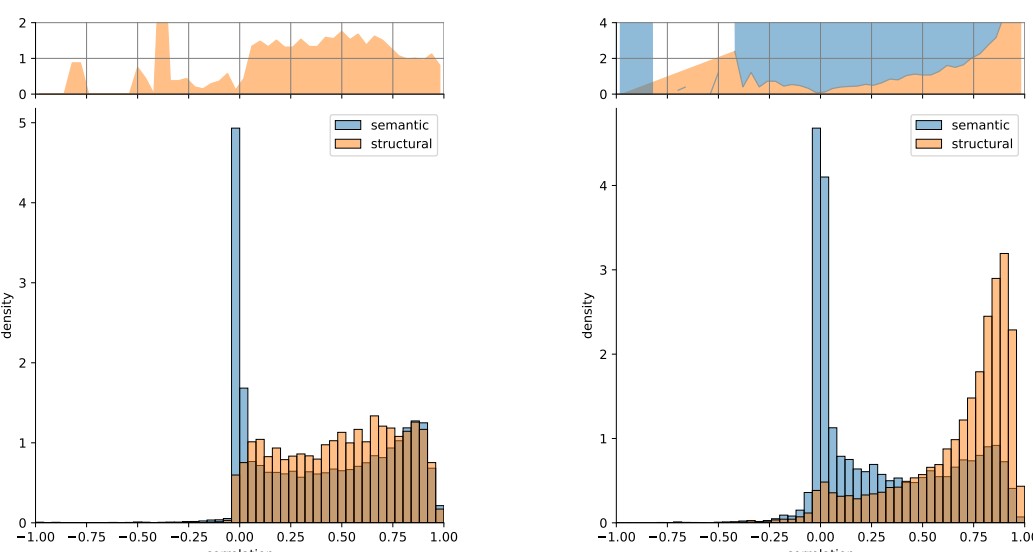

Figure 15: (Left) Pythia and (right) Gemma transfer differences between semantic and structural. We find somewhat similar patterns to the GPT2 case, namely the transferability of structural features and cluster of semantic features that do not transfer appears consistent. Since structural features naturally tend to be higher density, the stitch is incentivized via the MSE penalty to preserve those directions of space because there is higher variance in these dimensions and more consistently help reduce MSE.

```
- Ensure that the sentence remains grammatically correct, even though its
    meaning is completely different.
- Respond with the new sentence and nothing else.

Examples:
    Input: "The scientist carefully examined the ancient manuscript in the
        dimly lit library."
    Output: "The firefighter accidentally kicked the broken telephone in the
        noisy bus station."

    Input: "After the storm passed, the children ran outside to play in the
        puddles."
    Output: "Once the debate ended, the accountants flew abroad to invest in
        the factories."

Again, your instructions are:
```

Table 7: Examples of augmented sentences generated by our LLM procedure. Clearly there is some noise (a feature that only activates on "But" will be classified as structural) but we found these hard to avoid.

| Augmentation | Sentences |
|---|---|
| 1 | "But he added that Arizona's system also created problematic asymmetries and anomalies." |
| 2 | "But she sculpted that broccoli's texture also caused colorful duplicates and polygons." |
| 3 | "But she blended that broccoli also developed confusing melodies and constellations." |
| 4 | "But she designed that polka's rhythm also fabricated paradoxical similarities and galaxies." |
| 5 | "But he measured that broccoli's shape also produced complicated rhythms and symphonies." |
| 6 | "But she removed that penguin's melody also produced awkward polygons and tangents." |
| 1 | "Use promo code DOUG and play a real money game for FREE!" |
| 2 | "Bake discount coupon ZEBRA and practice a virtual cooking class for GOLD!" |
| 3 | "Order discount coupon ZETA and bake a frozen dinner for DINNER!" |
| 4 | "Use discount code ORANGE and bake a metal game for SILVER!" |
| 5 | "Apply offer code ZEBRA and bake a delicious casserole for DINNER!" |
| 6 | "Activate offer code ZEBRA and purchase a tropical fruit for GRATUITOUS!" |

Table 8: Examples of structural vs. semantic features based on feature centric evaluations. Drawn from `pythia-70m-deduped` pre-layer 3 SAEs. Tokens are highlighted based on activation strength.

| Feature Index | Category | Description | Selected Top Activating Examples |
|---|---|---|---|
| 18139 | Semantic | Geometric concepts | not passing through any vertex of a triangle
the concept of order into plane geometry |
| 4469 | Semantic | "-bility" words | increase brand awareness and desirability ahead
negative or every obstacle and impossibility |
| 889 | Structural | Nouns after "the" | of the confidentiality relating to the substance-abuse
the plot beats don't stray far from the genre |
| 11062 | Structural | Token after "and" list | Java, PHP, Python and Ruby still ensconced
solar, geothermal, hydroelectric, and biomass that |

```
- Ensure that the core meaning is entirely different from the original
    sentence.
- Do not use synonyms or words from the same semantic category.
- Maintain identical syntax, punctuation, and grammatical structure.
- The new sentence must be valid and natural, despite the unrelated meaning.

- Respond with the new sentence and nothing else.

Here is the original sentence.
Input: {sentence}
```

## H.2 Example Ablations and Classifications

See Table 7 for example generated augmentations and Table 8 for examples of classified features and their top activations.

# I Attention Patterns

See Figure 16 for an example of the attention deactivation acting as an attention activation feature for an attention head after being transferred.

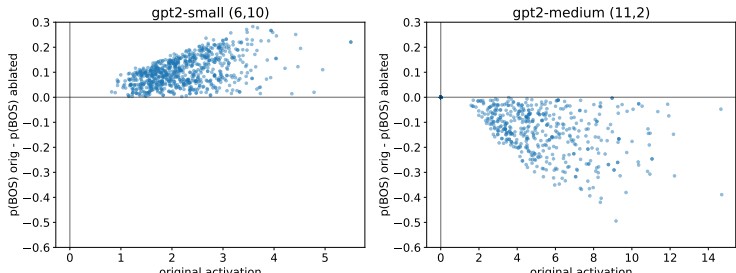

Figure 16: Example of how the same attention deactivation feature from before can act as an activator for a different downstream attention head after transfer.

## J  Compute Statement

All experiments were run on single Quadro 6000 / RTX3090 (both have 24GB VRAM) configurations. We estimate total GPU hours to be around 10000 in total for the project and most components of the experiments run contiguously for at most 1 day.

