# OpenReview forum: "Transferring Linear Features Across Language Models With Model Stitching"
_NeurIPS.cc/2025/Conference — NeurIPS 2025 spotlight_

### Official Review · Reviewer_jnC1 · 2025-06-06

**Clarity:** 3
**Significance:** 2
**Originality:** 2
**Rating:** 4
**Confidence:** 4

**Summary:**

The paper study the transferability of Sparse Autoencoders (SAE)'s features across different LLMs. Using a learned transformation, they stitch one LLMs features and inject them into another one, finding SAE shares universal features.
The work's main contribution is (i) analysis of features transferability and (ii) making the training of LLMs' SAE more efficient if there is a given and already trained SAE of a bigger LLM from the same model-family (GPT2,Pythia,Gemma).

**Questions:**

1. Is the geometric of trained SAE can already improve the initialization of new SAE, even without the stitching itself? (not a required experiments but a question I wonder about and can be useful in the limitation section).
2. Can your method reveal the gap between smaller and bigger models representation abilities?

**Ethical Concerns:**

["NO or VERY MINOR ethics concerns only"]

**Final Justification:**

I find the authors' answers interesting and I hope to see them incorporated into the paper in future versions.
I still find parts of the contribution to be limited (W2) but as an experimental setup I find it useful.
Overall, my concerns (W1-3) were acknowledged by the rebuttal and therefore I decided to increase my overall rating (3-->4).

**Limitations:**

yes

**Quality:**

3

**Strengths And Weaknesses:**

Strengths:
1. The paper is well written and easy to follow.
2. The idea of feature transferability is an interesting mechanistic problem, and the SAE's point of view as an innovative one (in particular in the cases of section 5, Feature Analysis).
3. The experiments are well conducted and the results are presented clearly.

Weaknesses:
1. Title and terminology: "features" has a wide use in deep learning. Since SAE are the center of the work, I would suggest incorporating "SAE" into the paper's title.
2. The contribution of initialization a new SAE from larger LLM's SAE is limited. While interesting as an experimental experiment, and even due the lower cost in FLOPs, it requires to train the stitching by its own, making the process more complex than just training the small SAE and highly depended on implementation (such as how we cache activation).
3. Applications' contribution limitations: section 4 discuss how stitching SAE's features of one LLMs can probe and steer another. In practice, the results have (i) only mediocre performances compared to native SAE's features, and (ii) amplifying the ability to stitch model's activation more than highlighting SAE usage in such cases. Therefore, I find this part to be more of an examination of the transferability levels between LLMs' SAE and less actual application.

---

> ### Author Rebuttal · Authors · 2025-07-31
>
> Thank you for your comments! We’re happy you found the paper easy to follow. On your concerns and questions:
> >**W1**: Title and terminology: "features" has a wide use in deep learning. Since SAE are the center of the work, I would suggest incorporating "SAE" into the paper's title.
>
> Although SAE features are a centerpiece of our study since they are the primary unsupervised method to extract large sets of features, we do still study features extracted not from SAEs, mainly in Section 4 where we also transfer probes trained on the residual stream and DiffMean steering vectors. Thus, we hesitate to center the title entirely around SAEs. That said, we appreciate the suggestion to be more specific with the loaded term “features” and will update the title to explicitly reflect our emphasis on linear features, or latents.
> >**W2**: The contribution of initialization a new SAE from larger LLM's SAE is limited. While interesting as an experimental experiment, and even due the lower cost in FLOPs, it requires to train the stitching by its own, making the process more complex than just training the small SAE and highly depended on implementation (such as how we cache activation).
>
> We agree that stitching and then transferring the SAE to get an initialization introduces some additional complexity compared to just training from scratch. However, the procedure was designed to be lightweight, as it is trained using simple MSE penalties, requires no architectural changes to the SAE, and activation caching is a standard step in SAE training. Given the significant FLOPs reduction, we believe this tradeoff is both practical and justified.
>
> >**W3**: Applications' contribution limitations: section 4 discuss how stitching SAE's features of one LLMs can probe and steer another. In practice, the results have (i) only mediocre performances compared to native SAE's features, and (ii) amplifying the ability to stitch model's activation more than highlighting SAE usage in such cases. Therefore, I find this part to be more of an examination of the transferability levels between LLMs' SAE and less actual application.
>
> While transferred features sometimes underperform native ones, we find that in several cases (e.g., specific probing datasets and some EuroParl languages), stitched vectors recover performance without any retraining of the probe/steering vector. These features are essentially “free,” requiring no additional optimization. We aim to explore the difference between scenarios where the transfer is successful vs. unsuccessful more deeply in future work.
>
> In response to your questions:
> > **Q1**: Is the geometric of trained SAE can already improve the initialization of new SAE, even without the stitching itself? (not a required experiments but a question I wonder about and can be useful in the limitation section).
>
> - This is a great question and touches on whether there exists a kind of universal “internal geometry” in the feature sets learned by SAEs. However, it's not clear to us how this could be directly exploited without stitching or identifying some other type of transformation, as the relationship between the geometries of different models may involve arbitrary transformations. For example, even if two models are “isomorphic” in some sense, a simple rotation or reflection might still be required to align their feature spaces. Perhaps an alternative approach could be used with similar methods as presented in [1] (see also, related work on multilingual transfer). It’s an interesting point we can spend more discussion on, but we leave this for future work.
>
> > **Q2**: Can your method reveal the gap between smaller and bigger models representation abilities?
>
> - We aim to address this question with our semantic/structural experiment. The preliminary conclusion we draw is that larger models have a set of semantic features they leave behind while mostly sharing structural features with the smaller model. A limitation however is that this distinction is quite high level, but to the best of our knowledge, no other method exists to get more specific insights in a systematic (i.e. no/minimal manual inspection) manner.
>
> We appreciate the reviewer's careful evaluation and hope these clarifications help demonstrate the practical value and generality of our approach. If these points have addressed your concerns, we would be grateful if you would consider raising your score, but please let us know if you have additional comments or concerns.
>
> ***References***
>
> [1] Li, Jiaang, et al. "Do vision and language models share concepts? a vector space alignment study." Transactions of the Association for Computational Linguistics 12 (2024): 1232-1249.

---

> > ### Comment · Reviewer_jnC1 · 2025-08-02
> >
> > I appreciate the detailed answer and I raise my overall rating. I will mention that I am interesting to see (in the future version of this work) the new experiment of Q2 and the extended discussion of Q1.

---

> > > ### Author Response · Authors · 2025-08-07
> > >
> > > Thank you for reading our response (and for raising your score)! We sincerely appreciate your feedback and will incorporate our discussions about Q1 and Q2 either in future versions of this work or follow up work.

---

### Official Review · Reviewer_2wwT · 2025-06-14

**Clarity:** 3
**Significance:** 2
**Originality:** 2
**Rating:** 5
**Confidence:** 4

**Summary:**

The paper proposes a method for transferring the weights of sparse autoencoders (SAEs) across different models belonging to the same family (i.e. same pre-training, same tokenizer but different size). This is especially interesting due to the possibility of bootstrapping SAEs for larger models by reusing those optimized for smaller versions of the same model. This translates to savings in terms of compute. Finally, the paper confirms advantages and limitations of their strategy on standard use cases/downstream tasks.

**Questions:**

I would kindly encourage the authors to refer to the Strengths/Weaknesses section. The only downside of this work is the limitation of being able to transfer SAEs only within the same model family. However at present, I see no other compelling reason to assign a lower score, and I remain open to revising my evaluation after engaging further with the authors' response and the other reviewers' feedbacks.

**Ethical Concerns:**

["NO or VERY MINOR ethics concerns only"]

**Final Justification:**

I would like to express my sincere appreciation to the authors for their thoughtful and comprehensive responses to the questions/concerns raised in my review. After careful consideration, and in light of the authors’ replies as well as the insights provided by the other reviewers, I raised my final score.

**Limitations:**

yes

**Quality:**

3

**Strengths And Weaknesses:**

The paper offers a view on what happens when transferring SAEs representations between models of different sizes. This complements the observations of [1]) and makes it an incremental (good) contribution on the phenomenon under study. Also, this has nice implications as the proposed technique translates to real-world savings in terms of costs. Finally, the use cases are convincing to confirm the effectivness of the approach and its limitations. I have very minor comments/questions:

----------------------

**Q1.**

Generally, when doing model stitching on large pre-trained models some low-capacity linear layer is sufficient to map features from one model to the other [2]. And it seems that your implementation agrees with such choice. So,
- What are the costs in terms of memory and runtime/FLOPs to optimize Eq.3?
- Could the four components of the loss in Eq.3 be broken down into four sub-problems that are solved in closed-form as regularized linear regression problems? Would it be a more cost-effective solution?

----------------------

**Q2.**

Given that one of the primary motivations and contributions of the paper revolves around computational efficiency, I would have expected a more fine-grained analysis on costs. As it stands, the only quantitative comparison I could locate is the FLOPs estimate reported in Appendix D. What about memory usage? (eg. Disk storage / RAM / VRAM)

----------------------

**Q3.**

I see a minor issue in the Introduction at L24-29, where the paper reports:

> The Linear Representation Hypothesis (LRH) posits that these features are represented as directions in a high dimensional space [Park et al., 2024, Elhage et al., 2022] and sparse dictionary learning methods [Olshausen and Field, 1996, Faruqui et al., 2015, Serre, 2006] such as Sparse Autoencoders (SAEs) have recently gained popularity for interpreting models through this lens [Huben et al., 2024, Bricken et al., 2023, Gao et al., 2025, Kissane et al., 2024].

I would argue that there is a missing link here between LRH and SAEs: sparse dictionary learning (eventually implemented with SAEs) has been introduced due to the problem of superposition of features given by polysemanticity [3]. I would suggest the authors to specify this explicitly.

----------------------

**_References:_**

[1] Connor Kissane, et al. "SAEs (usually) transfer between base and chat models." 2024. URL https://www.alignmentforum.org/posts/fmwk6qxrpW8d4jvbd/ saes-usually-transfer-between-base-and-chat-models.

[2] Bansal, Yamini, et al. "Revisiting model stitching to compare neural representations." NeurIPS 2021.

[3] Elhage, Nelson, et al. "Toy models of superposition." arXiv preprint arXiv:2209.10652 (2022).

---

> ### Author Rebuttal · Authors · 2025-07-31
>
> Thank you for the thoughtful feedback and we’re glad you found the contribution insightful. In response to your questions:
> > **Q1:** Generally, when doing model stitching on large pre-trained models some low-capacity linear layer is sufficient to map features from one model to the other [2]. And it seems that your implementation agrees with such choice. So, (1) What are the costs in terms of memory and runtime/FLOPs to optimize Eq.3? (2) Could the four components of the loss in Eq.3 be broken down into four sub-problems that are solved in closed-form as regularized linear regression problems? Would it be a more cost-effective solution?
>
> This is a great question and we also thought about this quite a bit in the beginning stages. As a baseline, we report the FLOPs numbers we used to generate our main results in Table 3 (namely, we found the stitch took ~$10^{15}$ FLOPs to train for about 200M tokens between `pythia-70m-deduped` and `pythia-160m-deduped`). There are three big points to make here.
> 1. Without the regularization terms, there are easy closed form solutions which can be computed in roughly $O(Td_1d_2)$ FLOPs. Using some napkin math, something on the order of $10^8$ to $10^9$ tokens can be used to fit the transformations and it would still be cheaper than our current gradient descent approach, which is more than enough. So indeed, it would be cheaper without the regularization terms.
> 2. However, including the inversion regularization terms results in *no* closed form, because the loss (which is MSE) now has quartic components (in $P_\uparrow$ and $P_\downarrow$) which won’t result in easily solvable forms when the gradient is 0. We noted in the Appendix A.2 that the inversion regularization terms helped with performant stitches, so we opted to include them.
> 3. Regardless, the FLOPs of the stitching procedure are essentially negligible compared to the cost of training the SAEs (10x less).
> We ended up settling with the regularized gradient descent approach to balance efficacy and simplicity of implementation, as we found the FLOPs of obtaining a stitch was always small relative to the cost of training the SAEs. Computing the closed form solutions would involve managing matrices which could potentially be very large (on the order of ~$10^9$ to $10^{10}$ elements even for just $10^6$ tokens) and could be a pain to set up and do efficiently (especially since taking the necessary matrix inverses could be unstable on lower precision).
>
> > **Q2:** Given that one of the primary motivations and contributions of the paper revolves around computational efficiency, I would have expected a more fine-grained analysis on costs. As it stands, the only quantitative comparison I could locate is the FLOPs estimate reported in Appendix D. What about memory usage? (eg. Disk storage / RAM / VRAM)
>
> A complete cost analysis is important, but it turns out not to differ between stitched initialization and random initialization in the mentioned categories. In our setup, disk usage for activation caching is identical across conditions: we trained the stitch for around 200M tokens over 2 epochs (i.e. 100M unique tokens), which would take ~250GB total @ fp16 to cache for both models. We note that importantly the activations used to train the stitch can be the same ones used to train the SAEs, so this doesn't actually mean one needs to cache another 250GB of distinct activations outside of those needed for training the SAEs. Furthermore, all SAEs share the same architecture, so there is no meaningful difference in RAM or VRAM usage. For this reason, we focused on FLOPs-to-convergence, which reflects the only nontrivial computational difference.
>
> > **Q3:** I see a minor issue in the Introduction at L24-29, where the paper reports...
>
> Thanks for pointing this out! We agree that connecting LRH and SAEs through the lens of superposition/polysemanticity would make the original motivation of SAEs clearer. We will include this change in future versions of the paper.
>
> Thank you again for your careful review and great questions. Please let us know if you have any other comments or concerns.

---

> > ### Comment · Reviewer_2wwT · 2025-08-05
> >
> > I would like to express my sincere appreciation to the authors for their thoughtful and comprehensive responses to the questions/concerns raised in my review. After careful consideration, and in light of the authors’ replies as well as the insights provided by the other reviewers, I raised my final score.

---

> > > ### Author Response · Authors · 2025-08-07
> > >
> > > Thank you for reading our response (and for raising your score)! We appreciate your positive feedback.

---

### Official Review · Reviewer_xWfH · 2025-07-01

**Clarity:** 3
**Significance:** 3
**Originality:** 2
**Rating:** 5
**Confidence:** 3

**Summary:**

The work trains a linear stitching layer on the residual streams of two models. They show that interpretability tools (SAEs, steering vectors) transfer under the stitching transformation (at least to some extent). The quality of this transfer is stress-tested under different use cases and shown to hold reasonably well. Moreover, they compute two projections, up and down, so that they are encouraged to be near inverses.

**Questions:**

- In the intialization, please provide the graph of the entire training graph of a representative example. Are there qualitative differences between the graphs of random and stitched training curves.

- Do you have intuition as to why FUV grows so much due to transfer? Is it just because of the richness of the larger model (I guess not, can be tested by taking the down projection of course), or is the transfer not clean with SAEs. And of course, is it solvable?

- You say in line 116, "transferred SAEs... are better than random". Is FUV for random close to 0.0. What if we init from the correct covariance of and mean?

- have you checked stitching between different architectures?

**Ethical Concerns:**

["NO or VERY MINOR ethics concerns only"]

**Final Justification:**

All relevant points were addressed in the original response. While I believe some points do need some further address, as discussed in the rebuttal response, these are minimal and won't influence my score.

**Limitations:**

As discussed above, mostly steering vector experiments lacking.

**Quality:**

3

**Strengths And Weaknesses:**

Strengths:
- Natural research question
- Comparison of several architecture families and several SAE architectures (though not combinations thereof)
- Good analysis.

** They separate the SAE features into two categories - semantic and structural, and check their respective transferabilities.

** Downstream preservation of the model's output is evaluated, and invertibility preservation is observed.

- Concrete application: They show that initialization of large SAE from small SAE can cut the training of the large SAE's training FLOPs by 2x.


Weaknesses:
- SAE initialization is tested only for a single SAE architecture
- Steering vector experiments are limited to EuroParl.

---

> ### Author Rebuttal · Authors · 2025-07-31
>
> We appreciate your comments and thank you for taking the time to review. We're glad you found the work well-motivated and clear. A couple of comments about the weaknesses mentioned:
> >**W1:** SAE initialization is tested only for a single SAE architecture.
>
> We only trained with one style of SAE (TopK). Our method for stitching is activation function agnostic, so this choice was mostly to remain consistent with state of the art SAEs and for simplicity of training: TopK SAEs are usually quite easy to train as opposed to JumpReLU, for example.
>
> >**W2:** Steering vector experiments are limited to EuroParl.
>
> We agree our experiments are limited in scope and note this in the Limitations. Our goal was to explore a broad range of applications within language models and as such prioritized general linear feature transfer. For another perspective on specifically the viability of transferring steering vectors, we recommend recent concurrent work such as [1] and [2].
>
> As for the questions:
> >**Q1:** In the intialization, please provide the graph of the entire training graph of a representative example. Are there qualitative differences between the graphs of random and stitched training curves.
>
> We include complete loss curves of the various SAEs we trained in Figure 1c (Pythia transfer) and Figure 8a (GPT2 transfer). Stitched initialization consistently reaches key thresholds faster than random initialization, which is where we estimated the ~50% FLOPs savings.
>
> >**Q2:** Do you have intuition as to why FUV grows so much due to transfer? Is it just because of the richness of the larger model (I guess not, can be tested by taking the down projection of course), or is the transfer not clean with SAEs. And of course, is it solvable?
>
> The main components explaining why the FUV increases after transfer are that (1) we are transferring from a low dimensional space ($\mathbb{R}^{d_A}$) to a higher dimensional space ($\mathbb{R}^{d_B}$), but the parameters of the transferred SAE are still rank $d_A \leq d_B$ and (2) richness of the larger model’s embedding space, as the reviewer correctly identified. There are probably other factors such as noise from the SAE features interacting with the transfer but we believe these are the two dominant sources.
>
> >**Q3:** You say in line 116, "transferred SAEs... are better than random". Is FUV for random close to 0.0. What if we init from the correct covariance of and mean?
>
> The FUV of the outputs of a randomly initialized SAE is not 0; we generally find it to be larger than 1 (worse than the mean of the activations over a dataset). As for initializing with mean and covariance, we're unclear how that would directly apply to SAE weights, which are learned sparse dictionaries rather than statistical estimators.
>
> >**Q4:**  have you checked stitching between different architectures?
>
> While we wanted to focus on just transferring between residual streams of language models and extracting as many applications as we could, prior work such as [3] explores cross-architecture and even cross-modality stitching, which we find complementary.
>
> Thank you again for your careful review and please let us know if you have any follow ups or other questions.
>
> ***References***
>
> [1] Lee, Andrew, et al. "Shared global and local geometry of language model embeddings." arXiv preprint arXiv:2503.21073 (2025).
>
> [2] Oozeer, Narmeen, et al. "Activation space interventions can be transferred between large language models." arXiv preprint arXiv:2503.04429 (2025).
>
> [3] Merullo, Jack, et al. "Linearly mapping from image to text space." arXiv preprint arXiv:2209.15162 (2022).

---

> ### Comment · Reviewer_xWfH · 2025-08-05
>
> The authors have addressed most of my concerns, and the score seems fair. Small problems still endure
> (1. mean and covariance are often relevant in model initialization, even in deep cases
> 2. I am familiar with Merullo et al.'s work, but this does not address the SAE transfer question, as stitching is already established, and not the point of this paper, the question is the impact on SAE), but I have no intention to be splitting hairs.

---

> > ### Author Response · Authors · 2025-08-07
> >
> > Thank you for reading our response! We sincerely appreciate your feedback.
> >
> > 1. There may be a way to incorporate the mean and covariance of the activations into SAE initialization, but to the best of our knowledge it's not immediately obvious how to do so nor is it common practice. We agree with the sentiment that other initialization measures could be compared against, though, and appreciate the comment.
> >
> > 2. Apologies, we interpreted the original question as referring to strictly stitching between different architectures. As for stitching and transferring SAEs, this is an interesting idea that we have not tried. Based on papers like Merullo et. al, we might expect that some natural language features would line up with visual counterparts if we transferred an SAE from a language model to a vision models.

---

### Official Review · Reviewer_CQm2 · 2025-07-04

**Clarity:** 3
**Significance:** 3
**Originality:** 3
**Rating:** 5
**Confidence:** 2

**Summary:**

The main idea of the paper is that affine transformations of the representations from the models of same family can be transferred to the another model of the same family, showing significant computational saving for training sparse autoencoders (SAEs).

The model stitching approach tried above although doesn't work for next-token prediction task, it is shown to be benefical for sparse auto-encoders training in the bigger model. The affine mappings are general enough to transfer other interpretable components like probes and steering vectors, which can effectively recover ground truth performance in various tasks.

A detailed analysis of feature transferability reveals that semantic and structural features behave differently during transfer, with structural features generally transferring more consistently, while semantic features are more polarized in their transfer success. The paper also identifies that the functional roles of specific features, such as entropy and attention deactivation features, are preserved after transfer

**Questions:**

1. To what extent does the dissimilarity in training datasets between the source and target language models affect the success and efficiency of feature transfer via model stitching? Have any experiments been conducted with models trained on significantly different data distributions?

2. Are there specific scenarios or model architectures where "model stitching" performs poorly or fails to effectively transfer features?

3. Can the model stitching be applied to individual attention heads or MLP layers, rather than solely relying on the residual streams?

**Ethical Concerns:**

["NO or VERY MINOR ethics concerns only"]

**Final Justification:**

I would like to keep my scores, as all my concerns are adequately answered.

**Limitations:**

Authors have adequately addressed the limitations.

**Paper Formatting Concerns:**

The paper reads well.

**Quality:**

3

**Strengths And Weaknesses:**

Strengths:

1. The idea of model stitching with its simple affine transformation for propagating information from smaller LLM to a larger LLM is interesting.

2. The proposed approach is useful for training SAEs faster on larger model with lesser flops.

3. The analysis on relatedness of the features across LLMs from same family is interesting.

Weakness:

1. While the paper focuses on affine mappings of residual streams, it may not capture more complex, non-linear relationships across models.

2. The methods works for the LLMs that share the same tokenization and same family. Curious to see how arbitrary LLMs with same tokenization can be stitched?

---

> ### Author Rebuttal · Authors · 2025-07-31
>
> Thank you for your thoughtful review! We’re glad you found the work interesting. Below, we address the limitations and questions raised.
> > **W1:** While the paper focuses on affine mappings of residual streams, it may not capture more complex, non-linear relationships across models.
>
> This is a great point - linear features were just the most convenient starting point for feature transfer. A natural follow-up question for future work is whether such mappings can even still preserve nonlinear structure, such as the circular features identified in [1].
>
> > **W2:** The methods works for the LLMs that share the same tokenization and same family. Curious to see how arbitrary LLMs with same tokenization can be stitched?
>
> We focused on models from the same family primarily due to simplicity and ease of access. In practice, it’s difficult to find arbitrary models from different families that use exactly the same tokenizer, which limited our ability to explore this further. If the reviewer is aware of such model pairs, we would be very interested in trying them. That said, we expect the method to generalize in principle to such cases, though possibly with slightly reduced performance due to differences in training distributions or architectural details.
>
> A few comments on the rest of the review:
> - **Contrary to the summary, model stitching does actually preserve next-token prediction loss**; this was our main benchmark for assessing stitch quality. We found it surprising that the residual streams of the model pairs could be so directly mapped.
>
> > **Q1:** To what extent does the dissimilarity in training datasets between the source and target language models affect the success and efficiency of feature transfer via model stitching? Have any experiments been conducted with models trained on significantly different data distributions?
>
> - We did not experiment with models trained on significantly different distributions, as our focus was on leveraging settings where we know the models were trained on highly similar distributions to transfer features. However, we agree this experiment could be a useful testbed for identifying whether stitching can pick up on the different features in the data distributions.
>
> >**Q2:** Are there specific scenarios or model architectures where "model stitching" performs poorly or fails to effectively transfer features?
>
> - There are indeed failure cases of feature transfer, as was the case with some languages in EuroParl. Another example was with the instruction following steering experiments, mentioned in Appendix F.2, in which we found that generally instruction following steering vectors uncovered using the DiffMean methodology proposed by [2] did not work as effectively after transfer. Our hypothesis is that the failures can be largely attributed to actual differences in representations between the models that emerge with scale (especially width of residual stream). This knowledge may be useful in understanding when we can expect different steering setups to work.
>
> > **Q3:** Can the model stitching be applied to individual attention heads or MLP layers, rather than solely relying on the residual streams?
>
> - While we focused on residual stream transfer, the same affine approach could extend to MLP or attention head transfer, likely with some slight adaptations. For MLPs, especially factual associations or interpretable functional directions (see Section 5.2), we’re optimistic. Attention heads are trickier because most attention heads are not well understood and can exhibit conditional behavior, so it is hard to predict how stitching would interact with the heads. We expect transfer to be more feasible when head behavior is well-characterized, such as induction heads or previous token heads. For example, one could consider stitching a fact from a large model to a small model, or similarly stitching an induction head circuit from a larger model to a smaller model that lacks an induction head circuit.
>
> We’re excited that the work spawns so many great follow-up questions and we hope this reflects well on its contribution. There is a lot of follow up work that we are excited to work on/see. Please let us know if you have any follow ups or other questions.
>
> ***References***
>
> [1] Joshua Engels, Eric J Michaud, Isaac Liao, Wes Gurnee, and Max Tegmark. Not all language model features are one-dimensionally linear. In The Thirteenth International Conference on Learning Representations, 2025b. URL https://openreview.net/forum?id=d63a4AM4hb.
>
> [2] Alessandro Stolfo, Vidhisha Balachandran, Safoora Yousefi, Eric Horvitz, and Besmira Nushi. Improving instruction-following in language models through activation steering. In The Thirteenth International Conference on Learning Representations, 2025. URL https://openreview.net/forum?id=wozhdnRCtw.

---

> > ### Comment · Reviewer_CQm2 · 2025-08-06
> >
> > The response was clear and addressed my questions. I have no further remarks and maintain my original evaluation.

---

> > > ### Author Response · Authors · 2025-08-07
> > >
> > > Thank you for reading our response! We sincerely appreciate your feedback.

---

### Decision · Program_Chairs · 2025-09-17

**Decision:**

Accept (spotlight)

**Comment:**

This paper presents a simple, effective, and practical method for transferring features between language models, with a clear and impactful application in making SAE training more efficient. The accompanying analyses provide valuable insights into the universality of representations across model scales. The work is well-executed, the claims are well-supported, and the paper sparked excellent discussion among the reviewers, who are in strong agreement on its quality. This is a solid contribution to the field.

All four reviewers recommend acceptance, with one being borderline accept. There's a clear consensus that the paper is well-written, technically solid, and addresses an interesting and important problem.

Strenghts:
* practical impact: all reviewers were impressed by the practical application accelerating SAE training (xWfH, 2wwT).
* novel analysis: The analysis of how different feature types transfer, perticulary the distinction between semantic and structural features.
* clarity and execution: the paper was praised for being clear, well-motivated, and easy to follow, with well-conducted experiments that support the claims (cQm2, jnC1)

Weaknesses:
The reviewers raised several constructive points, which authors addressed efficiently in their rebuttal:
* scope of experiment: CQm2 noted that the exp were limited to models with the same family and tokenizer. The authors acknowledged the limitations, explained the practical difficulties.
* cost analysis: 2wwT requested a more fine-grained cost analysis. the authors convincingly argued that the dominant memory cost.

The authors' rebuttal was thorough and well-receoved, successfully addressing the reviewers's concerns and leading to an increased from reviewer jnC1. Hence, i recommend accept.